# CALF: Benchmarking Evaluation of LFQA Using Chinese Examinations

## Abstract

Long-Form Question Answering (LFQA) refers to generating in-depth, paragraph-level responses to open-ended questions. Although lots of LFQA methods are developed, evaluating LFQA effectively and efficiently remains challenging due to its high complexity and cost. Therefore, there is no standard benchmark for LFQA evaluation till now. To address this gap, we make the first attempt by proposing a well-constructed, reference-based benchmark named **C**hinese ex**A**mination for **LFQ**A Evaluation (CALF), aiming to rigorously assess the performance of automatic evaluation metrics for LFQA. The CALF benchmark is derived from Chinese examination questions that have been translated into English. It includes up to 1476 examples consisting of knowledge-intensive and nuanced responses. Our evaluation comprises three different settings to analyze the behavior of automatic metrics comprehensively. We conducted extensive experiments on 7 traditional evaluation metrics, 3 prompt-based metrics, and 3 trained evaluation metrics, and tested on agent systems for the LFQA evaluation. The results reveal that none of the current automatic evaluation metrics shows comparable performances with humans, indicating that they cannot capture dense information contained in long-form responses well. In addition, we provide a detailed analysis of the reasons why automatic evaluation metrics fail when evaluating LFQA, offering valuable insights to advance LFQA evaluation systems.

## 1 Introduction

Long-form Question Answering (LFQA) (Fan et al., 2019) targets at generating in-depth, paragraph-level responses to open-ended questions. It requires models to have comprehensive domain-specific knowledge or use evidence from retrieved documents (Nakano et al., 2022; Akash et al., 2023) to provide accurate and relevant answers. Despite increased efforts have been put to enhance the reasonableness and completeness of long-form answers, developing automatic, reliable, and human-aligned evaluation metrics for LFQA is still unexplored. This oversight highlights the necessity for more robust approaches that can accurately evaluate the quality of generated responses in a way that aligns with human judgment and ensures trustworthiness.

However, evaluating LFQA is particularly challenging, as paragraph-level answers can overwhelm evaluators, requiring them to have a comprehensive understanding of the domain. Previous manual evaluations tend to rely on crowd-sourced workers for annotation, but the limited domain expertise inevitably causes low reliability. For automatic evaluation for LFQA, ROUGE (Lin, 2004) is always used to gauge the quality of the response relative to a reference. However, Krishna et al. (2021) have criticized ROUGE for its limited informativeness in long-form contexts. Recently, Xu et al. (2023) reveals these shortcomings and advocates for the use of experts to ensure higher annotation quality and test several automatic evaluation metrics according to expert annotation. Although they analyze how well automatic metrics align with expert preferences, a modest sample size and reliance on unstructured and reference-free ELI5 comments severely constrain their benchmark. Also, due to the advancement of Large Language Models (LLMs) (OpenAI, 2023; 2024), numerous studies begin to leverage LLMs to develop more nuanced evaluation metrics (Li et al., 2023; Jiang et al., 2024; Fan et al., 2024a;b). Therefore, the potential of LLMs in LFQA evaluation remains unexplored. Beyond the above efforts, our objective is to gain a comprehensive understanding of automatic evaluation methods for LFQA, encompassing both traditional metrics and LLM-based metrics.

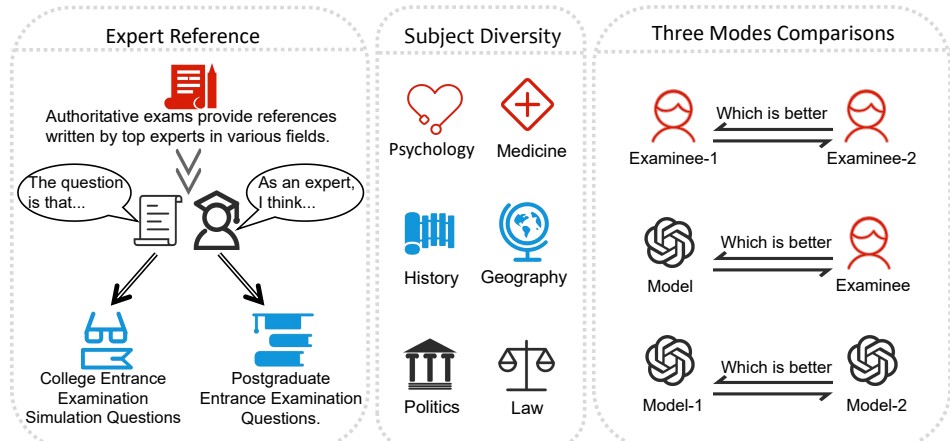

Figure 1: Features of CALF. Our CALF benchmark is built based on *Expert Reference*, *Subject Diversity*, and *Three Modes Comparisons*. We obtain *Expert Reference* by collecting data from Chinese Examinations, where professional teachers write references. To increase *Subject Diversity*, we source data from six subjects from more than 120 regional and national examinations. Besides, we experiment on *Human vs. Human*, *Human vs. Model*, and *Model vs. Model*. settings.

In this paper, we introduce the **C**hinese ex**A**mination for **L**ong **F**orm QA Evaluation Benchmark (CALF), comprising 1476 records with high-quality examination problems, responses for the problems (including human-written and model-generated), and meticulously crafted references. To ensure diversity and maintain quality, we source data from six domains—geography, history, politics, law, medicine, and psychology. They are primarily from the College Entrance Examination Simulation Questions (CEESQ) and the Postgraduate Entrance Examination Questions (PEEQ). These documents, initially in image format, are converted to plain text using OCR, with subsequent manual corrections to ensure accuracy. The overview of features of CALF is displayed in Figure 1.

For geography, history, and politics, we capture context, questions, references, and student responses, while for law, medicine, and psychology, we collect only context, questions, and references since we have no access to student responses. Following the methodologies of Xu et al. (2022), considering that these references are displayed in a concise and structured form for the teachers to use, we transform these references into more natural and long-form narratives to bridge the gap between structured and natural discourse.

We use automatic evaluation metrics to compare two responses and select the better one (Xu et al., 2023) or give a "tie" option. To ensure a comprehensive evaluation, we establish three comparison modes: *human vs. human*, *human vs. model*, and *model vs. model* for history, politics, and geography subjects, and *model vs. model* exclusively for law, medicine, and psychology subjects as we have no access to student responses which means that the two responses to be evaluated are either model-generated or human-written. Model-generated responses are produced from Llama-3-8b (Dubey et al., 2024) and GPT-3.5-turbo-1106-preview (OpenAI, 2023). Human-written responses are collected from online exams. Due to the responses of humans are knowledge-intensive and nuanced, and the responses from models are lengthy and rich in knowledge, they are effective for testing the evaluation of LFQA by identifying information in a given text and comparing to select a better one. Expert annotators from relevant domains are enlisted to participate in the annotation process, ensuring that the assessments are grounded in domain-specific knowledge.

Using CALF, we critically assess the efficacy of traditional, prompt-only, and trained automatic evaluation metrics against human judgment. Our experimental findings indicate that no current automatic metrics achieve parity with human preferences. Also, LLMs are not suitable metrics for LFQA evaluation using the vanilla prompt, Chain of Thoughts, and G-Eval. We also display agreement rates between automatic evaluation metrics and provide further error analysis to figure out why current evaluation metrics cannot handle LFQA evaluation. Our results reveal that both traditional evaluation metrics and LLM-based evaluation metrics cannot exclude the auxiliary information away from the key information and thus are insensitive to the minor differences in the given comparison pairs. Therefore, utilizing CoT or majority voting offers minimal improvement to the evaluation process. At last, we conduct experiments with agents for LFQA evaluation. Suprisingly, the results

| Dataset | Num. Dataset | Length # of Words | | | | | Source |
|---|---|---|---|---|---|---|---|
| | | Avg. Q | Avg. R | Max. R | Avg. A | Max. A | |
| Politics | **396** | **23.7** | 391.0 | 624 | **161.9** | 459 | CEESQ |
| History | 392 | 22.9 | 326.7 | **746** | 147.8 | **469** | CEESQ |
| Geography | 392 | 17.4 | 97.8 | 209 | 128.4 | 386 | CEESQ |
| Law | 100 | 10.7 | **469.7** | 642 | 151.3 | 394 | PEEQ |
| Psychology | 96 | 10.6 | 426.7 | 718 | 151.4 | **469** | PEEQ |
| Medicine | 100 | 8.8 | 322.9 | 492 | **161.9** | 374 | PEEQ |

Table 1: Statistics of our CALF benchmark. Q, R and A represent questions, references and responses respectively. CEESQ is College Entrance Examination Simulation Questions while PEEQ stands for Postgraduate Entrance Examination Questions. The maximum values are denoted using **bold**.

reveal that compared with a single model, agent systems do not show significant performance improvements. The findings guide our exploration of methods to enhance LFQA evaluation metrics, calling for a novel approach that captures nuanced semantic differences between responses clearly and correctly, pointing the way toward future advancements in the field.

## 2 RELATED WORK

**Development of LFQA**    LFQA (Fan et al., 2019) requires models to generate paragraph-level responses to open-ended questions which is more complex compared to datasets like SQuAD (Rajpurkar et al., 2016), TriviaQA (Joshi et al., 2017), and NarrativeQA (Kočiský et al., 2017), where answers are primarily words or phrases extracted directly from documents. In LFQA, models must generate nuanced responses based on their knowledge or existing evidence documents. Several studies have analyzed the discourse structure of long-form answers (Xu et al., 2022) and have sought to enhance the performance on LFQA. (Chen et al., 2023a; Akash et al., 2023).

**Evaluation of LFQA**    The automatic evaluation of LFQA remains challenging and underexplored. Initially, ROUGE (Lin, 2004) was used as an automatic evaluation metric to calculate the similarity between a candidate and a reference. Later, Krishna et al. (2021) pointed out that ROUGE is not an adequately informative metric for LFQA evaluation. For human annotation, HURDLES (Krishna et al., 2021) and WEBGPT (Nakano et al., 2022) employed A/B testing, where crowdsourced annotators were instructed to choose the better of two candidate answers. Since annotation of LFQA requires high expertise, the results of crowdsourced workers may be unreliable. To address the gap, Xu et al. (2023) employed experts for annotation, and tested several evaluation metrics, such as ROUGE (Lin, 2004), BERTScore (Zhang et al., 2020), and BARTScore (Yuan et al., 2021), on an expert-annotated dataset. Their findings validated that no existing metrics fully align with human judgment. However, the dataset they used lacks expert-written references, sourced from Reddit/ELI5, and is limited in scale, comprising only about 120 samples. In response to these limitations, we propose constructing a high-quality and larger benchmark for the evaluation of LFQA.

## 3 METHODOLOGY

To reasonably test the evaluation ability of different metrics for LFQA when having a reference to look up, we construct CALF, a comprehensive benchmark composed of different topics and questions. In this section, we will describe the construction process of CALF in detail.

### 3.1 DESIGN PRINCIPLE

**Reference-Based Evaluation**    The evaluation of LFQA systems critically depends on the availability of high-quality reference answers. These references serve as benchmarks, allowing evaluators to determine the precision and completeness of responses generated by the QA systems. Specifically, a reference answer provides a baseline to assess whether the response captures all necessary details and adheres closely to the factual correctness established in the reference. This dual check ensures that the generated answer is not only correct but also exhaustive in covering the question's scope.

However, crafting such references is both challenging and resource-intensive. Producing a reliable reference requires domain experts who have a vast knowledge and understanding of the subject

| Dataset | Human vs Human | Human vs Model | Model vs Model |
|---------|----------------|----------------|----------------|
| Politics | 0.80 | 0.87 | 0.76 |
| History | 0.74 | 0.77 | 0.72 |
| Geography | 0.75 | 0.80 | 0.68 |
| Law | N/A | N/A | 0.74 |
| Psychology | N/A | N/A | 0.75 |
| Medicine | N/A | N/A | 0.72 |

Table 2: Inter-annotator agreement of our annotation process. For politics, history, and geography, we have collected human-written answers and thereby annotated them under three settings. For law, psychology, and medicine, we only have model-generated answers, so we leave out *Human vs. Human* and *Human vs. Model* settings.

matter and thus is rather expensive, as it demands significant time and compensation for these professionals to generate each reference. In our settings, references are sourced from academic and professional examinations, where they are crafted by educational specialists.

**Authoritative Data** For our benchmark, we source our data from the College Entrance Examination Simulation Questions (CEESQ) and Postgraduate Entrance Examination Questions (PEEQ). The former pertains to regional examinations, while the latter is associated with national examinations. We avoid using questions from the actual College Entrance Examination to prevent overlap with potential training data, ensuring the integrity and independence of our evaluation dataset.

Questions for both CEESQ and PEEQ are meticulously crafted by domain experts who aim to test students' understanding and reasoning capabilities. The reference answers accompanying these questions are well-curated, containing all essential information pertinent to the questions. Additionally, we enhance our dataset with student responses collected from the corresponding Online Marking Platforms (OMP). These responses are typically more nuanced and knowledge-intensive compared to those found in ELI5, reflecting the rigorous academic standards and detailed content required in examination settings.

**Diverse Benchmark** To guarantee a diverse and representative benchmark for our LFQA system, we meticulously collect questions in six distinct domains: history, politics, geography, law, medicine, and psychology. For each domain, we draw from a pool of over 20 examination papers, ensuring a broad spectrum of topics and difficulty levels.

Our data in history, politics, and geography are sourced from CEESQ. Here, we have access to student responses via the OMP. From this platform, we select pairs of student responses that are particularly clear and detailed, enhancing the quality of our dataset. In addition, we employ LLMs to generate model responses, enabling us to rigorously test and compare the evaluation capabilities of various metrics between human-generated and model-generated responses. For the domains of law, medicine, and psychology, we do not use real student responses, but rely solely on model-generated answer pairs. This approach allows us to focus on assessing the performance of our LFQA system in generating high-quality, accurate responses in highly specialized and technical fields.

### 3.2 Overview

CALF consists of 1476 examples from Geography, Politics, History, Law, Medicine, and Psychology. The statistics are listed in Table 1. We also categorize the records into seven groups. Detailed explanations and examples are displayed in Appendix A.1 and Appendix A.2.

### 3.3 Data Processing

The data processing pipeline can be divided into three phases: **Data Collection**, **Data Transformation**, and **Model Response Generation**.

**Data Collection** We gather examination papers primarily in image format and employ Optical Character Recognition (OCR) systems to meticulously extract various components such as contexts,

questions, student responses, and references. For the fields of Law, Medicine, and Psychology, where student responses are inaccessible, we focus exclusively on extracting contexts, questions, and references. The OCR system is conducted using the API provided by VolcEngine. After OCR processing, we conduct a thorough manual review to correct any recognition errors.

**Data Transformation** References, when sourced from examination settings, are meticulously structured into concise information points, a format that facilitates scoring for educators. However, unlike these examination-derived references, those collected from evidence documents in practical applications often take the form of natural language narratives, incorporating examples, summaries, and other auxiliary information (Xu et al., 2022). To bridge this gap and enhance the utility of our references, we employ GPT-4o (OpenAI, 2024) to enrich our concise references with detailed explanations and expanded arguments. We list detailed explanations and human annotations of our reference transformation process in Appendix B.2. In addition, we provide an example in Appendix B.3. The transformation prompt is shown in Appendix C.7.

**Model Response Generation** To rigorously assess the performance of LFQA systems, our benchmark employs three comparison modes for each question, i.e., *human vs. human*, *human vs. model*, and *model vs. model* in Geography, History, and Politics, and *model vs. model* exclusively in Law, Medicine, and Psychology. That is, the two responses to be compared are either human-written or model-generated. When generating model responses, we focus on evaluating whether LLMs can understand the semantic meaning of texts well and properly select the better response. Therefore, we do not impose strict requirements on answer quality. Instead, we ensure the difficulty of CALF by selecting models with similar ranking in the LMSYS Arena (Chiang et al., 2024; Zheng et al., 2023; 2024b). Specifically, we leverage Llama-3-8B (Dubey et al., 2024) and GPT-3.5-turbo-1106-preview (OpenAI, 2023) for response generation. For model-generated answers, we prompt using "Generate reasonable answers to the following questions. Use references or examples if needed", and the generation temperature is set to 1.0 to encourage diverse and creative responses.

### 3.4 HUMAN ANNOTATION

The Human Annotation Process can be separated into the following steps: **Annotator Decision**, **Annotation Setting**, **Annotation Process**, and **After-Annotation Validation**.

**Annotator Decision** LFQA evaluation suffer from distinct challenges. Firstly, paragraph-level responses can overwhelm annotators, leading to a loss of focus. Secondly, annotators must have deep domain knowledge to accurately judge responses against references. Lastly, the syntactic and semantic complexities of long-form responses often intertwine correct and incorrect information within single sentences. To address these issues, we first hire five annotators from relevant aspects or who have taken relevant courses, including Computer Science, Law, and Medicine. Then we provide them with clear and detailed annotation recipes for better quality control.

**Annotation Setting** Guided by Xu et al. (2023), our evaluation criteria mainly focus on factuality, completeness, and clarity according to the reference. Unlike typical A/B testing, our method employs a triple-choice format to better capture the subtle differences between answers, as they often show comparable levels of information overlap with the reference, with additional information useless or verbose according to the central topic.

**Annotation Process** The annotators assess two responses against a given reference and select the more informative and complete answer or declare a "tie" if both are comparable. The process includes **Identify Key Information**, **Check for Key Information in Responses**, **Handling Responses**, and **Compare Overlapping Information**. During the process, we treat a piece of information as the basic unit. Firstly, annotators extract the key information needed to answer the question and check whether the responses under evaluation contain similar statements. If a similar one is present, they further assess whether this statement is fully correct, partially correct, or entirely incorrect. Finally, they will select a better one based on the overlapped information. Our annotation guidelines are provided in Appendix B.1. Detailed explanations and examples of fully correct, partially correct, and entirely incorrect answers can be found in Appendix B.1 and Appendix B.3.

| Metric | Geography | History | Politics | Psychology | Medicine | Law | Avg. |
|---|---|---|---|---|---|---|---|
| Traditional Metrics | | | | | | | |
| ROUGE | 0.482 | 0.513 | 0.417 | 0.448 | **0.630** | 0.360 | 0.471 |
| BLEU | 0.349 | 0.372 | 0.386 | 0.479 | 0.530 | 0.399 | 0.425 |
| BERTScore | 0.429 | 0.421 | 0.379 | 0.438 | 0.510 | 0.460 | 0.440 |
| BLEURT | 0.508 | 0.474 | 0.480 | 0.427 | 0.430 | 0.505 | 0.466 |
| BARTScore | 0.490 | 0.508 | 0.505 | 0.479 | 0.600 | 0.510 | 0.515 |
| UniEval | **0.533** | 0.503 | 0.381 | 0.323 | 0.570 | 0.560 | 0.470 |
| GPT2♠ | 0.429 | 0.490 | 0.467 | 0.427 | 0.610 | 0.590 | 0.502 |
| Prompt-only Metrics | | | | | | | |
| ChatGPT | 0.510 | 0.533 | 0.495 | 0.531 | 0.590 | 0.520 | 0.525 |
| ChatGPT-CoT | 0.505 | 0.518 | **0.530** | 0.521 | 0.610 | 0.520 | 0.531 |
| G-Eval (ChatGPT) | 0.508 | 0.526 | 0.520 | 0.521 | 0.600 | 0.610 | 0.545 |
| GPT-4o | 0.490 | 0.513 | 0.516 | 0.510 | 0.580 | 0.590 | 0.532 |
| GPT-4o-CoT | 0.503 | **0.561** | 0.523 | **0.563** | 0.620 | **0.620** | **0.563** |
| G-Eval (GPT-4o) | 0.482 | 0.548 | 0.522 | 0.531 | **0.630** | 0.480 | 0.531 |
| Trained Metrics | | | | | | | |
| Critique-6B | 0.462 | 0.528 | 0.424 | 0.330 | 0.340 | 0.460 | 0.400 |
| AutoJ-13B | 0.446 | 0.508 | 0.480 | 0.500 | 0.580 | 0.580 | 0.517 |
| TIGERScore-13B♠ | 0.406 | 0.268 | 0.212 | 0.282 | 0.320 | 0.320 | 0.295 |

Table 3: Performance of Evaluation Metrics. The baselines denoted by ♠ are reference-free metrics. ChatGPT used here is the GPT-3.5-turbo-1106-preview version. The results are measured by the matching rate of model preference and human preference in the triple-choice settings. The largest accuracy rate is denoted using **bold**. For traditional evaluation metrics, we round their output values to two place decimals to ensure the option "tie" will happen. The results indicate the inferior capability of current automatic evaluation metrics to capture and understand key information in a given text. The results indicate that no evaluation metrics show comparable results with humans.

**After-Annotation Validation**  To minimize bias and subjectivity, each record is annotated by two independent reviewers. The inter-annotator agreement is reported in Table 2. We annotate politics, history, and geography in three settings and law, psychology, and medicine in only the *Model vs. Model* setting. The inter-annotator agreement indicates that the agreement rate is highest when comparing human responses with model responses. However, the overall inter-annotator agreement highlight the challenges of LFQA evaluation, as none of them exceed 90%. When disagreements occur, a third annotator is asked to make the final decision, and justification is added if necessary. We provide case studies for annotation and justification in Appendix B.3 to help understand.

## 4 EXPERIMENTS

We experiment using several automatic evaluation metrics on CALF benchmark. We report their performance and provide detailed analysis on the evaluation of LFQA on CALF.

### 4.1 EXPERIMENTAL SETUP

#### 4.1.1 BASELINES

To provide a thorough overview of evaluation metrics on LFQA, we test three kinds of metrics.

**Traditional evaluation metrics**  We test several general-purpose evaluation metrics including ROUGE-L (Lin, 2004), BLEU (Papineni et al., 2002), BERTScore (Zhang et al., 2020), BARTScore (Yuan et al., 2021), GPT-2 Perplexity, UniEval (Zhong et al., 2022), and BLEURT (Sellam et al., 2020). Since these metrics are based on returned values that can hardly be the same, we round these values to two decimal places to make the option "tie" happen.

**Prompt-only evaluation metrics**  We select GPT-3.5-turbo-1106-preview (OpenAI, 2023) and GPT-4o (OpenAI, 2024) as our evaluators, because they are always considered one of the most capable LLMs and they are used as backbones for many tasks. Besides, we implement CoT (Wei

| Metric | Geography | | | Politics | | | History | | |
|---|---|---|---|---|---|---|---|---|---|
| | H/H | H/M | M/M | H/H | H/M | M/M | H/H | H/M | M/M |
| Traditional Metrics | | | | | | | | | |
| ROUGE | 0.510 | 0.530 | 0.357 | 0.485 | 0.399 | 0.383 | 0.439 | 0.505 | **0.602** |
| BLEU | 0.459 | 0.357 | 0.224 | 0.424 | 0.384 | 0.354 | 0.021 | 0.474 | 0.520 |
| BERTScore | 0.479 | 0.479 | 0.336 | 0.424 | 0.338 | 0.414 | 0.378 | 0.469 | 0.367 |
| BLEURT | 0.622 | **0.551** | 0.306 | 0.515 | 0.500 | 0.404 | 0.480 | 0.495 | 0.429 |
| BARTScore | 0.551 | 0.479 | **0.449** | 0.600 | 0.490 | 0.444 | 0.510 | 0.505 | 0.510 |
| UniEval | 0.612 | 0.525 | 0.469 | 0.323 | 0.455 | 0.293 | 0.531 | 0.500 | 0.480 |
| GPT2♠ | 0.489 | 0.413 | 0.398 | 0.556 | 0.434 | 0.444 | 0.469 | 0.474 | 0.541 |
| Prompt-only Metrics | | | | | | | | | |
| ChatGPT | 0.653 | 0.494 | 0.397 | 0.589 | 0.500 | 0.404 | 0.551 | 0.515 | 0.551 |
| ChatGPT-CoT | 0.693 | 0.505 | 0.316 | 0.626 | **0.525** | 0.440 | 0.561 | 0.485 | 0.541 |
| G-Eval (ChatGPT) | 0.653 | 0.500 | 0.377 | 0.626 | 0.495 | 0.465 | 0.551 | 0.520 | 0.515 |
| GPT-4o | 0.653 | 0.479 | 0.346 | **0.677** | 0.455 | **0.475** | 0.571 | 0.480 | 0.520 |
| GPT-4o-CoT | 0.663 | 0.489 | 0.367 | 0.626 | 0.495 | **0.475** | **0.622** | **0.536** | 0.551 |
| G-Eval (GPT-4o) | **0.704** | 0.474 | 0.275 | 0.626 | 0.500 | 0.465 | 0.582 | 0.531 | 0.551 |
| Trained Metrics | | | | | | | | | |
| Critique-6B | 0.612 | 0.489 | 0.255 | 0.465 | 0.455 | 0.323 | 0.459 | 0.500 | 0.571 |
| AutoJ-13B | 0.561 | 0.469 | 0.285 | 0.586 | 0.490 | 0.444 | 0.510 | 0.515 | 0.571 |
| TIGERScore-13B♠ | 0.449 | 0.392 | 0.387 | 0.171 | 0.202 | 0.272 | 0.367 | 0.306 | 0.092 |

Table 4: Detailed Evaluation Metrics Across Different Domains. The baselines denoted by ♠ are reference-free evaluation metrics. H/H stands for *Human vs Human*, H/M stands for *Human vs Model*, and M/M stands for *Model vs Model*. ChatGPT used here is the GPT-3.5-turbo-1106-preview version. The matching rate of model preference and human preference in the triple-choice settings measures all the results. The largest accuracy rate is denoted using **bold**. For traditional evaluation metrics, we round their output values to two place decimals to ensure the option "tie" will happen.

et al., 2023) to prompt them to think step by step and get the final answer. Also, we implement G-Eval (Liu et al., 2023b) with the number of responses equal to 5. Since we only care about the better one in two responses, we simplify the weighted summation in G-Eval using Majority Voting.

**Trained evaluation metrics** Metrics tested here are not trained for LFQA specially, but for NLG. We assume that the metrics for the evaluation of NLG can be transferred to the evaluation of LFQA. We select Auto-J-13B(Li et al., 2023), TIGERScore-13B(Jiang et al., 2024), and CritiqueLLM-6B (Ke et al., 2024). The detailed description of trained metrics above is listed in Appendix C.1.

### 4.1.2 TASK SET

Firstly, We translate the dataset into English using GPT-4o (OpenAI, 2024) with the prompt "Translate the following text into ENGLISH" with temperature equals 1.0 to ensure a fair comparison since some of the models cannot achieve comparable results in Chinese. To mitigate the influence of the order of responses displayed on subsequent analysis for using LLMs for evaluation (Pezeshkpour & Hruschka, 2023), we balance the rate of preference by alternating the sequence of the answers. As we mentioned in Section 3.4, we set the evaluation as a triple choice according to a reference between two responses, generated either by humans or models. For traditional evaluation metrics, we use the provided references as references and the responses as hypotheses for the metrics. Then we compare the scores of the two responses to make the decision. For close-sourced models, we adopt a 0-shot setting with a direct prompt to finish the triple-choice task. For open-sourced models, we modify the data format following their instructions. Since they have structured output, we use regex to find the corresponding parts and obtain our results. For all of the LLM-based evaluation metrics, we use temperatures equal to 1.0. The prompt for LLMs is shown in Appendix C.7.

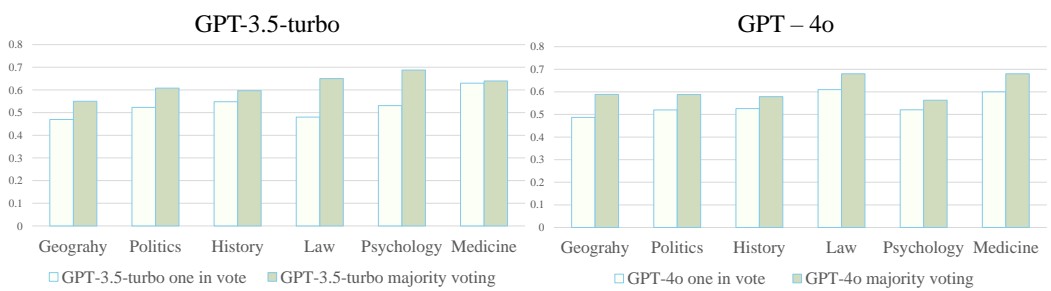

Figure 2: Comparison of *One in Votes* and *Majority Voting* where *One in Votes* stands for whether the gold label exists in the votes and *Majority Voting* stands for the most votes equal to the gold label. The left is the comparison in the ChatGPT setting and the right is in the GPT-4o setting.

## 4.2 EXPERIMENTAL RESULTS

The overall experimental results are shown in Table 3. We also report our results without tied QA pairs in Appendix C.2.

**CALF as a Challenging Benchmark**   The CALF benchmark poses significant challenges for all existing automatic evaluation metrics for LFQA, with accuracy rates consistently falling below 65%, suggesting performance levels not much better than random guessing. All traditional metrics are not suitable for LFQA evaluation due to their static property, thus failing to handle flexible outputs. Even advanced models like GPT-4o struggle to consistently make correct evaluations, though it remains the most promising among the available automatic metrics.

To provide a more detailed analysis, we report the accuracy rates for three comparison modes—*human vs. human*, *human vs. model*, and *model vs. model*—within the domains of geography, politics, and history, as shown in Table 4.

The results from these detailed experiments reveal that a majority of evaluation metrics perform best in the *human vs. human* setting and fail in the *Model vs. Model* setting and *Human vs. Model* setting. We hypothesize that human-written answers tend to be more precise and concise, making it knowledge-intensive and easier for the evaluation metrics to distinguish the better answer (Saito et al., 2023). Furthermore, there is a notable variation in performance across different domains, particularly with LLM-based metrics. For instance, GPT-4o achieves an accuracy of only 35% in Geography compared to 59% in Law at the *model vs model* setting.

**CoT and G-Eval Provide Limited Improvement for LLMs in LFQA Evaluation**   To implement CoT, we prompt the LLMs to "think step by step." This method generally yields better performance compared to vanilla prompts. However, the observed performance gains are not substantial, suggesting that CoT alone is insufficient for significantly enhancing LFQA evaluation.

Additionally, we employ G-Eval by generating responses from LLMs five times and selecting the most frequent answer. The results show that G-Eval does not consistently improve performance, further suggesting that LLMs may struggle to robustly determine the better answer (Zheng et al., 2024a). This underlines the need for more advanced techniques to enhance the reliability and accuracy of LFQA evaluations.

To further assess the effectiveness of G-Eval (Liu et al., 2023b), we analyze the consistency of the model's responses across five iterations. The results, presented in Figure 2, suggest that by sampling multiple times, the model can make correct decisions, but the consistency of these decisions across iterations is low. This indicates that while the model has the potential to be used for LFQA evaluation, its ability to consistently choose the correct answer with higher reliability—or "confidence"—needs to be improved.

**To What Extent Do Automatic Metrics Agree with Each Other?**   We analyze the agreement rates between pairs of automatic evaluation metrics, including a baseline called the "always long response" option, which consistently selects the longer answer. While longer responses may seem more informative, this is not always the case. We show our results in Figure 3. indicate that LLM-based evaluation metrics tend to correlate highly with each other and exhibit a slight preference for

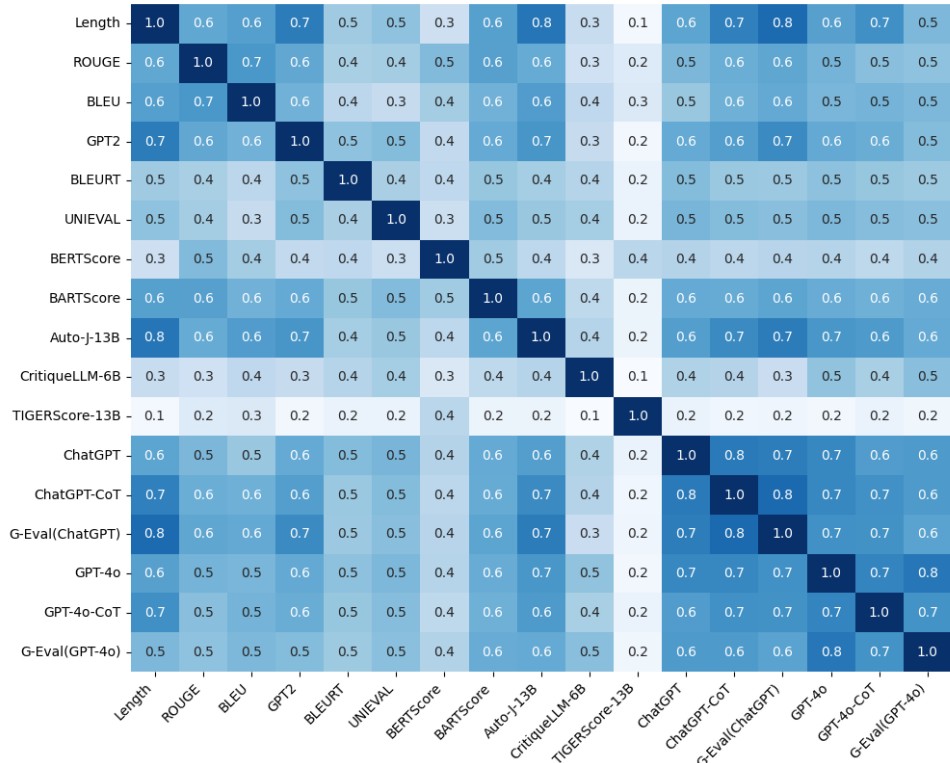

Figure 3: Overall correlation between automatic evaluation baselines. The darker color indicates a larger correlation. The annotation on each block is calculated by the ratio of the intersection between $BaselineA$ and $BaselineB$ and the total number of records tested by both $BaselineA$ and $BaselineB$.

length-oriented metrics. In contrast, traditional evaluation metrics show lower agreement and are more likely to produce divergent results. To have a deeper analysis, we list the detailed results under three modes separately in Appendix C.3.

**Do Automatic Evaluation Metrics Mirror Human Challenges?**   We investigate two scenarios: i). cases where there is disagreement among human annotators, but the evaluation metrics align with the meta-review, ii). cases where human annotators agree, and the metrics make the correct choice. The experimental results detailed in Appendix C.4 reveal that automatic evaluation metrics do not fully replicate the challenges faced by humans. However, even when human annotators reach unanimous agreement, the accuracy of automatic metrics is lower, suggesting that these models have a reduced capacity to capture semantic nuances effectively.

### 4.3 ERROR ANALYSIS

To provide a comprehensive understanding of why automatic evaluation metrics fall short, we present an error analysis in Appendix C.5. This analysis explores the specific reasons behind the failures of LLM-based and traditional evaluation metrics when compared to human evaluation. The results indicate that traditional metrics primarily fail due to their inherent static nature, making them unsuitable for the flexible and diverse outputs. On the other hand, LLM-based metrics struggle because they often fail to capture the key points well, leading to unfair or inaccurate evaluations.

## 5 CAN LLM AGENTS HELP EVALUATION OF LFQA?

Given the growing trend of leveraging agent for complex reasoning tasks (Yao et al., 2023; Liu et al., 2023a; Chen et al., 2023b), their potential use for evaluation has garnered attention in recent research (Chan et al., 2023; Narsupalli et al., 2024). In this section, we aim to investigate whether agents can

| Metric | Geography | History | Politics | Psychology | Medicine | Law | Avg. |
|---|---|---|---|---|---|---|---|
| ChatGPT-ICL | 0.482 | **0.528** | **0.507** | 0.500 | 0.630 | **0.570** | **0.538** |
| GPT-4o-ICL | 0.497 | **0.528** | **0.507** | **0.531** | 0.610 | 0.520 | 0.533 |
| ChatEval | **0.523** | **0.528** | 0.503 | 0.417 | 0.590 | **0.570** | 0.530 |
| ReFeR | 0.474 | 0.461 | 0.500 | 0.500 | **0.660** | 0.540 | 0.523 |

Table 5: Results of evaluation metrics including ICL metrics and Agent-based Metrics. The results are measured by the matching rate of model preference of human preference. The largest rate is denoted using **bold**.

be effectively utilized for LFQA evaluation. We select ChatGPT-ICL and GPT-4o-ICL as our baselines with three demonstrations and test on CHATEVAL (Chan et al., 2023), and REFER (Narsupalli et al., 2024) on our benchmark and list our results in Table 5. The detailed implementation of our agent system is listed in Appendix C.6. The prompts for agent systems are shown in Appendix C.7.

**In-Context Learning Fails to Improve Performance on CALF** We tested LLM-based evaluation metrics using three examples, but the results showed no significant improvement, with some subjects even experiencing a notable performance drop. This underscores the challenges posed by CALF and the inherent difficulty of LFQA evaluation. Additionally, GPT-4o, despite having more parameters, only achieved results comparable to ChatGPT, suggesting that larger models do not necessarily enhance performance in LFQA evaluation.

**The Performance of Agent Systems for Evaluation Requires Further Exploration** We tested two SOTA agent systems using our CALF benchmark. The experimental results indicate that neither system consistently outperforms simple LLMs, although there are instances where evaluation performance improved for specific subjects. This may be due to hallucinations in one step causing a cascading effect on subsequent steps, without the opportunity for correction. The optimal structure and model settings for agent systems in the context of evaluation require further investigation to unlock their full potential.

# 6 DISCUSSION AND FUTURE WORK

Here, we discuss the automatic evaluation of LFQA and the feature work in this area.

**Evaluation of LFQA Remains Challenging** Despite the growing use of LLMs for evaluation tasks, our findings demonstrate that relying solely on LLMs for LFQA evaluation is challenging and unreliable. Also, traditional metrics also fall short in this context. Therefore, the exploration of semantic-sensitive evaluation metrics is essential for accurate LFQA assessment. Our benchmark, built on expert-written questions and references, lays a solid foundation for future research in developing more effective evaluation methods.

**Inclusion of Multi-Disciplines** In this work, we primarily focus on plain text records within the realm of social sciences, as we believe that most LFQA tasks originate in this format and domain. However, recognizing that some LFQA questions may pertain to natural sciences, we plan to expand our dataset to include subjects like mathematics and physics. By incorporating question types such as theorem explanations, we aim to enhance the diversity and comprehensiveness of our benchmark, making it more authoritative and representative across multiple disciplines.

# 7 CONCLUSION

We present CALF, a comprehensive benchmark consisting of up to 1,500 records from six subjects, sourced from various Chinese examinations. This benchmark is designed to evaluate the effectiveness of current LFQA evaluation metrics including traditional metrics, prompt-based metrics, and trained metrics, and shed light on the evaluation of LFQA. Additionally, we conduct experiments using agent-system evaluation metrics. Our future work will focus on developing semantic-sensitive evaluation metrics and expanding the dataset to include a broader range of disciplines.

## AUTHOR CONTRIBUTIONS

If you'd like to, you may include a section for author contributions as is done in many journals. This is optional and at the discretion of the authors.

## ACKNOWLEDGMENTS

Use unnumbered third-level headings for the acknowledgments. All acknowledgments, including those to funding agencies, go at the end of the paper.

## REFERENCES

Pritom Saha Akash, Kashob Kumar Roy, Lucian Popa, and Kevin Chen-Chuan Chang. Long-form question answering: An iterative planning-retrieval-generation approach, 2023. URL https://arxiv.org/abs/2311.09383.

Chi-Min Chan, Weize Chen, Yusheng Su, Jianxuan Yu, Wei Xue, Shanghang Zhang, Jie Fu, and Zhiyuan Liu. Chateval: Towards better llm-based evaluators through multi-agent debate, 2023. URL https://arxiv.org/abs/2308.07201.

Hung-Ting Chen, Fangyuan Xu, Shane Arora, and Eunsol Choi. Understanding retrieval augmentation for long-form question answering, 2023a. URL https://arxiv.org/abs/2310.12150.

Weize Chen, Yusheng Su, Jingwei Zuo, Cheng Yang, Chenfei Yuan, Chi-Min Chan, Heyang Yu, Yaxi Lu, Yi-Hsin Hung, Chen Qian, Yujia Qin, Xin Cong, Ruobing Xie, Zhiyuan Liu, Maosong Sun, and Jie Zhou. Agentverse: Facilitating multi-agent collaboration and exploring emergent behaviors, 2023b. URL https://arxiv.org/abs/2308.10848.

Wei-Lin Chiang, Lianmin Zheng, Ying Sheng, Anastasios Nikolas Angelopoulos, Tianle Li, Dacheng Li, Hao Zhang, Banghua Zhu, Michael Jordan, Joseph E. Gonzalez, and Ion Stoica. Chatbot arena: An open platform for evaluating llms by human preference, 2024.

Abhimanyu Dubey, Abhinav Jauhri, Abhinav Pandey, Abhishek Kadian, Ahmad Al-Dahle, Aiesha Letman, Akhil Mathur, Alan Schelten, Amy Yang, Angela Fan, Anirudh Goyal, Anthony Hartshorn, Aobo Yang, Archi Mitra, Archie Sravankumar, Artem Korenev, Arthur Hinsvark, Arun Rao, Aston Zhang, Aurelien Rodriguez, Austen Gregerson, Ava Spataru, Baptiste Roziere, Bethany Biron, Binh Tang, Bobbie Chern, Charlotte Caucheteux, Chaya Nayak, Chloe Bi, Chris Marra, Chris McConnell, Christian Keller, Christophe Touret, Chunyang Wu, Corinne Wong, Cristian Canton Ferrer, Cyrus Nikolaidis, Damien Allonsius, Daniel Song, Danielle Pintz, Danny Livshits, David Esiobu, Dhruv Choudhary, Dhruv Mahajan, Diego Garcia-Olano, Diego Perino, Dieuwke Hupkes, Egor Lakomkin, Ehab AlBadawy, Elina Lobanova, Emily Dinan, Eric Michael Smith, Filip Radenovic, Frank Zhang, Gabriel Synnaeve, Gabrielle Lee, Georgia Lewis Anderson, Graeme Nail, Gregoire Mialon, Guan Pang, Guillem Cucurell, Hailey Nguyen, Hannah Korevaar, Hu Xu, Hugo Touvron, Iliyan Zarov, Imanol Arrieta Ibarra, Isabel Kloumann, Ishan Misra, Ivan Evtimov, Jade Copet, Jaewon Lee, Jan Geffert, Jana Vranes, Jason Park, Jay Mahadeokar, Jeet Shah, Jelmer van der Linde, Jennifer Billock, Jenny Hong, Jenya Lee, Jeremy Fu, Jianfeng Chi, Jianyu Huang, Jiawen Liu, Jie Wang, Jiecao Yu, Joanna Bitton, Joe Spisak, Jongsoo Park, Joseph Rocca, Joshua Johnstun, Joshua Saxe, Junteng Jia, Kalyan Vasuden Alwala, Kartikeya Upasani, Kate Plawiak, Ke Li, Kenneth Heafield, Kevin Stone, Khalid El-Arini, Krithika Iyer, Kshitiz Malik, Kuenley Chiu, Kunal Bhalla, Lauren Rantala-Yeary, Laurens van der Maaten, Lawrence Chen, Liang Tan, Liz Jenkins, Louis Martin, Lovish Madaan, Lubo Malo, Lukas Blecher, Lukas Landzaat, Luke de Oliveira, Madeline Muzzi, Mahesh Pasupuleti, Mannat Singh, Manohar Paluri, Marcin Kardas, Mathew Oldham, Mathieu Rita, Maya Pavlova, Melanie Kambadur, Mike Lewis, Min Si, Mitesh Kumar Singh, Mona Hassan, Naman Goyal, Narjes Torabi, Nikolay Bashlykov, Nikolay Bogoychev, Niladri Chatterji, Olivier Duchenne, Onur Çelebi, Patrick Alrassy, Pengchuan Zhang, Pengwei Li, Petar Vasic, Peter Weng, Prajjwal Bhargava, Pratik Dubal, Praveen Krishnan, Punit Singh Koura, Puxin Xu, Qing He, Qingxiao Dong, Ragavan Srinivasan, Raj Ganapathy, Ramon Calderer, Ricardo Silveira Cabral, Robert Stojnic, Roberta Raileanu, Rohit Girdhar, Rohit Patel, Romain Sauvestre, Ronnie Polidoro, Roshan Sumbaly, Ross Taylor, Ruan Silva, Rui Hou, Rui Wang, Saghar Hosseini, Sahana Chennabasappa,

Sanjay Singh, Sean Bell, Seohyun Sonia Kim, Sergey Edunov, Shaoliang Nie, Sharan Narang, Sharath Raparthy, Sheng Shen, Shengye Wan, Shruti Bhosale, Shun Zhang, Simon Vandenhende, Soumya Batra, Spencer Whitman, Sten Sootla, Stephane Collot, Suchin Gururangan, Sydney Borodinsky, Tamar Herman, Tara Fowler, Tarek Sheasha, Thomas Georgiou, Thomas Scialom, Tobias Speckbacher, Todor Mihaylov, Tong Xiao, Ujjwal Karn, Vedanuj Goswami, Vibhor Gupta, Vignesh Ramanathan, Viktor Kerkez, Vincent Gonguet, Virginie Do, Vish Vogeti, Vladan Petrovic, Weiwei Chu, Wenhan Xiong, Wenyin Fu, Whitney Meers, Xavier Martinet, Xiaodong Wang, Xiaoqing Ellen Tan, Xinfeng Xie, Xuchao Jia, Xuewei Wang, Yaelle Goldschlag, Yashesh Gaur, Yasmine Babaei, Yi Wen, Yiwen Song, Yuchen Zhang, Yue Li, Yuning Mao, Zacharie Delpierre Coudert, Zheng Yan, Zhengxing Chen, Zoe Papakipos, Aaditya Singh, Aaron Grattafiori, Abha Jain, Adam Kelsey, Adam Shajnfeld, Adithya Gangidi, Adolfo Victoria, Ahuva Goldstand, Ajay Menon, Ajay Sharma, Alex Boesenberg, Alex Vaughan, Alexei Baevski, Allie Feinstein, Amanda Kallet, Amit Sangani, Anam Yunus, Andrei Lupu, Andres Alvarado, Andrew Caples, Andrew Gu, Andrew Ho, Andrew Poulton, Andrew Ryan, Ankit Ramchandani, Annie Franco, Aparajita Saraf, Arkabandhu Chowdhury, Ashley Gabriel, Ashwin Bharambe, Assaf Eisenman, Azadeh Yazdan, Beau James, Ben Maurer, Benjamin Leonhardi, Bernie Huang, Beth Loyd, Beto De Paola, Bhargavi Paranjape, Bing Liu, Bo Wu, Boyu Ni, Braden Hancock, Bram Wasti, Brandon Spence, Brani Stojkovic, Brian Gamido, Britt Montalvo, Carl Parker, Carly Burton, Catalina Mejia, Changhan Wang, Changkyu Kim, Chao Zhou, Chester Hu, Ching-Hsiang Chu, Chris Cai, Chris Tindal, Christoph Feichtenhofer, Damon Civin, Dana Beaty, Daniel Kreymer, Daniel Li, Danny Wyatt, David Adkins, David Xu, Davide Testuggine, Delia David, Devi Parikh, Diana Liskovich, Didem Foss, Dingkang Wang, Duc Le, Dustin Holland, Edward Dowling, Eissa Jamil, Elaine Montgomery, Eleonora Presani, Emily Hahn, Emily Wood, Erik Brinkman, Esteban Arcaute, Evan Dunbar, Evan Smothers, Fei Sun, Felix Kreuk, Feng Tian, Firat Ozgenel, Francesco Caggioni, Francisco Guzmán, Frank Kanayet, Frank Seide, Gabriela Medina Florez, Gabriella Schwarz, Gada Badeer, Georgia Swee, Gil Halpern, Govind Thattai, Grant Herman, Grigory Sizov, Guangyi, Zhang, Guna Lakshminarayanan, Hamid Shojanazeri, Han Zou, Hannah Wang, Hanwen Zha, Haroun Habeeb, Harrison Rudolph, Helen Suk, Henry Aspegren, Hunter Goldman, Igor Molybog, Igor Tufanov, Irina-Elena Veliche, Itai Gat, Jake Weissman, James Geboski, James Kohli, Japhet Asher, Jean-Baptiste Gaya, Jeff Marcus, Jeff Tang, Jennifer Chan, Jenny Zhen, Jeremy Reizenstein, Jeremy Teboul, Jessica Zhong, Jian Jin, Jingyi Yang, Joe Cummings, Jon Carvill, Jon Shepard, Jonathan McPhie, Jonathan Torres, Josh Ginsburg, Junjie Wang, Kai Wu, Kam Hou U, Karan Saxena, Karthik Prasad, Kartikay Khandelwal, Katayoun Zand, Kathy Matosich, Kaushik Veeraraghavan, Kelly Michelena, Keqian Li, Kun Huang, Kunal Chawla, Kushal Lakhotia, Kyle Huang, Lailin Chen, Lakshya Garg, Lavender A, Leandro Silva, Lee Bell, Lei Zhang, Liangpeng Guo, Licheng Yu, Liron Moshkovich, Luca Wehrstedt, Madian Khabsa, Manav Avalani, Manish Bhatt, Maria Tsimpoukelli, Martynas Mankus, Matan Hasson, Matthew Lennie, Matthias Reso, Maxim Groshev, Maxim Naumov, Maya Lathi, Meghan Keneally, Michael L. Seltzer, Michal Valko, Michelle Restrepo, Mihir Patel, Mik Vyatskov, Mikayel Samvelyan, Mike Clark, Mike Macey, Mike Wang, Miquel Jubert Hermoso, Mo Metanat, Mohammad Rastegari, Munish Bansal, Nandhini Santhanam, Natascha Parks, Natasha White, Navyata Bawa, Nayan Singhal, Nick Egebo, Nicolas Usunier, Nikolay Pavlovich Laptev, Ning Dong, Ning Zhang, Norman Cheng, Oleg Chernoguz, Olivia Hart, Omkar Salpekar, Ozlem Kalinli, Parkin Kent, Parth Parekh, Paul Saab, Pavan Balaji, Pedro Rittner, Philip Bontrager, Pierre Roux, Piotr Dollar, Polina Zvyagina, Prashant Ratanchandani, Pritish Yuvraj, Qian Liang, Rachad Alao, Rachel Rodriguez, Rafi Ayub, Raghotham Murthy, Raghu Nayani, Rahul Mitra, Raymond Li, Rebekkah Hogan, Robin Battey, Rocky Wang, Rohan Maheswari, Russ Howes, Ruty Rinott, Sai Jayesh Bondu, Samyak Datta, Sara Chugh, Sara Hunt, Sargun Dhillon, Sasha Sidorov, Satadru Pan, Saurabh Verma, Seiji Yamamoto, Sharadh Ramaswamy, Shaun Lindsay, Shaun Lindsay, Sheng Feng, Shenghao Lin, Shengxin Cindy Zha, Shiva Shankar, Shuqiang Zhang, Shuqiang Zhang, Sinong Wang, Sneha Agarwal, Soji Sajuyigbe, Soumith Chintala, Stephanie Max, Stephen Chen, Steve Kehoe, Steve Satterfield, Sudarshan Govindaprasad, Sumit Gupta, Sungmin Cho, Sunny Virk, Suraj Subramanian, Sy Choudhury, Sydney Goldman, Tal Remez, Tamar Glaser, Tamara Best, Thilo Kohler, Thomas Robinson, Tianhe Li, Tianjun Zhang, Tim Matthews, Timothy Chou, Tzook Shaked, Varun Vontimitta, Victoria Ajayi, Victoria Montanez, Vijai Mohan, Vinay Satish Kumar, Vishal Mangla, Vlad Ionescu, Vlad Poenaru, Vlad Tiberiu Mihailescu, Vladimir Ivanov, Wei Li, Wenchen Wang, Wenwen Jiang, Wes Bouaziz, Will Constable, Xiaocheng Tang, Xiaofang Wang, Xiaojian Wu, Xiaolan Wang, Xide Xia, Xilun Wu, Xinbo Gao, Yanjun Chen, Ye Hu, Ye Jia, Ye Qi, Yenda Li, Yilin Zhang, Ying Zhang, Yossi Adi, Youngjin

Nam, Yu, Wang, Yuchen Hao, Yundi Qian, Yuzi He, Zach Rait, Zachary DeVito, Zef Rosnbrick, Zhaoduo Wen, Zhenyu Yang, and Zhiwei Zhao. The llama 3 herd of models, 2024. URL https://arxiv.org/abs/2407.21783.

Angela Fan, Yacine Jernite, Ethan Perez, David Grangier, Jason Weston, and Michael Auli. Eli5: Long form question answering, 2019. URL https://arxiv.org/abs/1907.09190.

Yuchen Fan, Yantao Liu, Zijun Yao, Jifan Yu, Lei Hou, and Juanzi Li. Evaluating generative language models in information extraction as subjective question correction, 2024a. URL https://arxiv.org/abs/2404.03532.

Yuchen Fan, Xin Zhong, Chengsi Wang, Gaoche Wu, and Bowen Zhou. Eva-score: Evaluation of long-form summarization on informativeness through extraction and validation, 2024b. URL https://arxiv.org/abs/2407.04969.

Dongfu Jiang, Yishan Li, Ge Zhang, Wenhao Huang, Bill Yuchen Lin, and Wenhu Chen. Tigerscore: Towards building explainable metric for all text generation tasks, 2024. URL https://arxiv.org/abs/2310.00752.

Mandar Joshi, Eunsol Choi, Daniel S. Weld, and Luke Zettlemoyer. Triviaqa: A large scale distantly supervised challenge dataset for reading comprehension, 2017. URL https://arxiv.org/abs/1705.03551.

Pei Ke, Bosi Wen, Zhuoer Feng, Xiao Liu, Xuanyu Lei, Jiale Cheng, Shengyuan Wang, Aohan Zeng, Yuxiao Dong, Hongning Wang, Jie Tang, and Minlie Huang. Critiquellm: Towards an informative critique generation model for evaluation of large language model generation, 2024. URL https://arxiv.org/abs/2311.18702.

Tomáš Kočiský, Jonathan Schwarz, Phil Blunsom, Chris Dyer, Karl Moritz Hermann, Gábor Melis, and Edward Grefenstette. The narrativeqa reading comprehension challenge, 2017. URL https://arxiv.org/abs/1712.07040.

Kalpesh Krishna, Aurko Roy, and Mohit Iyyer. Hurdles to progress in long-form question answering, 2021. URL https://arxiv.org/abs/2103.06332.

Junlong Li, Shichao Sun, Weizhe Yuan, Run-Ze Fan, Hai Zhao, and Pengfei Liu. Generative judge for evaluating alignment, 2023. URL https://arxiv.org/abs/2310.05470.

Chin-Yew Lin. ROUGE: A package for automatic evaluation of summaries. In *Text Summarization Branches Out*, pp. 74–81, Barcelona, Spain, July 2004. Association for Computational Linguistics. URL https://aclanthology.org/W04-1013.

Bo Liu, Yuqian Jiang, Xiaohan Zhang, Qiang Liu, Shiqi Zhang, Joydeep Biswas, and Peter Stone. Llm+p: Empowering large language models with optimal planning proficiency, 2023a. URL https://arxiv.org/abs/2304.11477.

Yang Liu, Dan Iter, Yichong Xu, Shuohang Wang, Ruochen Xu, and Chenguang Zhu. G-eval: Nlg evaluation using gpt-4 with better human alignment, 2023b. URL https://arxiv.org/abs/2303.16634.

Reiichiro Nakano, Jacob Hilton, Suchir Balaji, Jeff Wu, Long Ouyang, Christina Kim, Christopher Hesse, Shantanu Jain, Vineet Kosaraju, William Saunders, Xu Jiang, Karl Cobbe, Tyna Eloundou, Gretchen Krueger, Kevin Button, Matthew Knight, Benjamin Chess, and John Schulman. Webgpt: Browser-assisted question-answering with human feedback, 2022. URL https://arxiv.org/abs/2112.09332.

Yaswanth Narsupalli, Abhranil Chandra, Sreevatsa Muppirala, Manish Gupta, and Pawan Goyal. Review-feedback-reason (refer): A novel framework for nlg evaluation and reasoning, 2024. URL https://arxiv.org/abs/2407.12877.

OpenAI. Chatgpt: Chat generative pre-trained transformer. https://chat.openai.com/, 2023. Accessed: 2024-08-05.

OpenAI. Hello gpt-4o. https://openai.com/index/hello-gpt-4o/, 2024. Accessed: 2024-08-05.

Kishore Papineni, Salim Roukos, Todd Ward, and Wei-Jing Zhu. Bleu: a method for automatic evaluation of machine translation. In Pierre Isabelle, Eugene Charniak, and Dekang Lin (eds.), *Proceedings of the 40th Annual Meeting of the Association for Computational Linguistics*, pp. 311–318, Philadelphia, Pennsylvania, USA, July 2002. Association for Computational Linguistics. doi: 10.3115/1073083.1073135. URL https://aclanthology.org/P02-1040.

Pouya Pezeshkpour and Estevam Hruschka. Large language models sensitivity to the order of options in multiple-choice questions, 2023. URL https://arxiv.org/abs/2308.11483.

Pranav Rajpurkar, Jian Zhang, Konstantin Lopyrev, and Percy Liang. Squad: 100,000+ questions for machine comprehension of text, 2016. URL https://arxiv.org/abs/1606.05250.

Keita Saito, Akifumi Wachi, Koki Wataoka, and Youhei Akimoto. Verbosity bias in preference labeling by large language models. *ArXiv*, abs/2310.10076, 2023. URL https://api.semanticscholar.org/CorpusID:264147087.

Thibault Sellam, Dipanjan Das, and Ankur P. Parikh. Bleurt: Learning robust metrics for text generation, 2020. URL https://arxiv.org/abs/2004.04696.

Jason Wei, Xuezhi Wang, Dale Schuurmans, Maarten Bosma, Brian Ichter, Fei Xia, Ed Chi, Quoc Le, and Denny Zhou. Chain-of-thought prompting elicits reasoning in large language models, 2023. URL https://arxiv.org/abs/2201.11903.

Fangyuan Xu, Junyi Jessy Li, and Eunsol Choi. How do we answer complex questions: Discourse structure of long-form answers, 2022. URL https://arxiv.org/abs/2203.11048.

Fangyuan Xu, Yixiao Song, Mohit Iyyer, and Eunsol Choi. A critical evaluation of evaluations for long-form question answering, 2023. URL https://arxiv.org/abs/2305.18201.

Shunyu Yao, Jeffrey Zhao, Dian Yu, Nan Du, Izhak Shafran, Karthik Narasimhan, and Yuan Cao. React: Synergizing reasoning and acting in language models, 2023. URL https://arxiv.org/abs/2210.03629.

Weizhe Yuan, Graham Neubig, and Pengfei Liu. Bartscore: Evaluating generated text as text generation, 2021. URL https://arxiv.org/abs/2106.11520.

Tianyi Zhang, Varsha Kishore, Felix Wu, Kilian Q. Weinberger, and Yoav Artzi. Bertscore: Evaluating text generation with bert, 2020. URL https://arxiv.org/abs/1904.09675.

Chujie Zheng, Hao Zhou, Fandong Meng, Jie Zhou, and Minlie Huang. Large language models are not robust multiple choice selectors, 2024a. URL https://arxiv.org/abs/2309.03882.

Lianmin Zheng, Wei-Lin Chiang, Ying Sheng, Siyuan Zhuang, Zhanghao Wu, Yonghao Zhuang, Zi Lin, Zhuohan Li, Dacheng Li, Eric Xing, Hao Zhang, Joseph E. Gonzalez, and Ion Stoica. Judging llm-as-a-judge with mt-bench and chatbot arena. In *Thirty-seventh Conference on Neural Information Processing Systems Datasets and Benchmarks Track*, 2023. URL https://openreview.net/forum?id=uccHPGDlao.

Lianmin Zheng, Wei-Lin Chiang, Ying Sheng, Tianle Li, Siyuan Zhuang, Zhanghao Wu, Yonghao Zhuang, Zhuohan Li, Zi Lin, Eric Xing, Joseph E. Gonzalez, Ion Stoica, and Hao Zhang. Lmsys-chat-1m: A large-scale real-world llm conversation dataset. In *The Twelfth International Conference on Learning Representations*, 2024b. URL https://openreview.net/forum?id=BOfDKxfwt0.

Ming Zhong, Yang Liu, Da Yin, Yuning Mao, Yizhu Jiao, Pengfei Liu, Chenguang Zhu, Heng Ji, and Jiawei Han. Towards a unified multi-dimensional evaluator for text generation, 2022. URL https://arxiv.org/abs/2210.07197.

## A  BENCHMARK DETAILS

### A.1  QUESTION CATEGORY

We categorize our benchmark into seven groups: Factoid Questions, Comparative Questions, Definition Questions, Analytical Questions, Evaluation Questions, Inferential Questions, and Application Questions. The detailed ratios of each question type are presented in Figure 4.

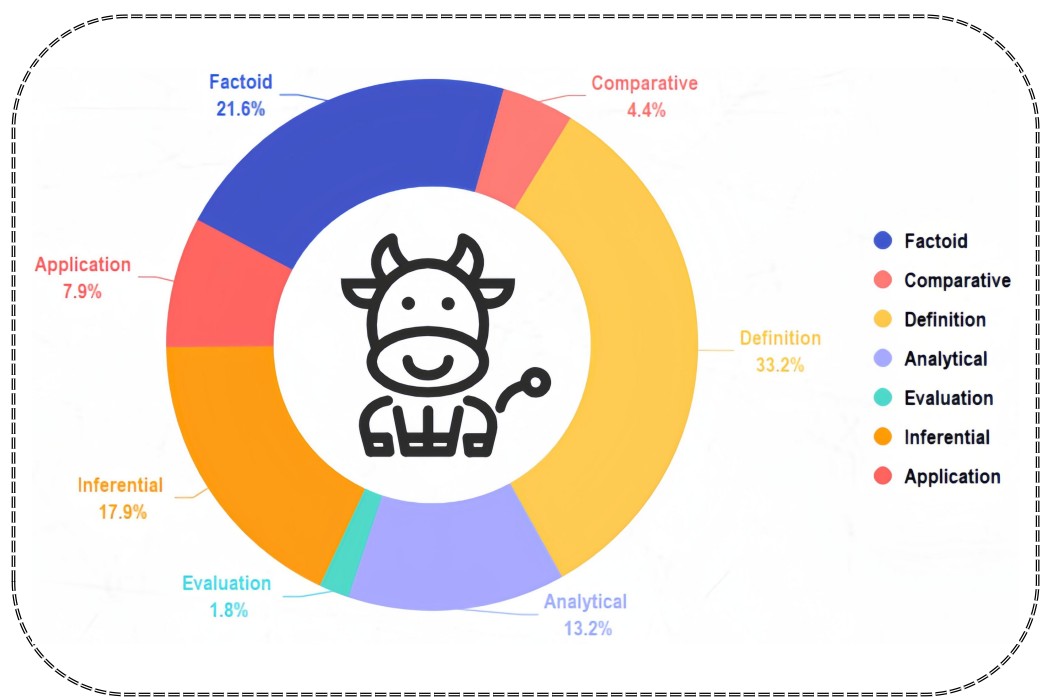

Figure 4: The ratios of each question type in our CALF benchmark, including Factoid Questions, Comparative Questions, Definition Questions, Analytical Questions, Evaluation Questions, Inferential Questions, and Application Questions.

### A.2  QUESTION DEFINITIONS AND EXAMPLES

In this section, we provide definitions and examples for each kind of question mentioned above including Factoid Questions, Comparative Questions, Definition Questions, Analytical Questions, Evaluation Questions, Inferential Questions, and Application Questions.

**Factoid Questions**   Factoid Questions, in the context of Long-Form Question Answering (LFQA), demand comprehensive, paragraph-level responses to specific factual queries, rather than the brief Yes/No answers typically associated with factoid queries. These questions are designed to assess a model's ability to not only retrieve and present accurate factual information but also to articulate this information in a coherent, well-structured narrative. The example is shown in Table 7.

**Comparative Questions**   Comparative Questions involve analyzing the similarities and differences between two or more entities, events, or concepts. These questions prompt the responder to draw comparisons and highlight distinct features or characteristics. The example is shown in Table 8.

**Definition Questions**   Definition Questions require a clear explanation or clarification of a specific term, concept, or object. These questions are focused on ensuring that the respondent accurately understands and can articulate the meaning of particular terminology. The example is shown in Table 9.

**Analytical Questions** Analytical Questions require breaking down a complex topic or scenario into its fundamental components to understand the underlying reasons, mechanisms, or causes. These questions often involve a deep examination of how different factors interact. The example is shown in Table 10.

**Evaluation Questions** Evaluation Questions ask the responder to assess or judge the value, quality, or impact of a particular concept, event, or action. These questions require considering different perspectives and making informed decisions or arguments. The example is shown in Table 11

**Inferential Questions** Inferential Questions require drawing conclusions or making deductions based on the provided information and prior knowledge. These questions often involve interpreting data or scenarios to reach a logical outcome or understanding. The example is shown in Table 12.

**Application Questions** Application Questions involve applying theoretical knowledge, concepts, or principles to real-world scenarios or specific situations. These questions test the ability to use learned material in practical contexts. The example is shown in Table 13

## B  ANNOTATION DETAILS

### B.1  ANNOTATION RECIPE

**Annotation Process:**  When provided with a question, a reference, and two candidate answers, the evaluation process should proceed as follows:

1. **Identify Key Information:** We begin by listing all the key pieces of information required to answer the question based on the reference. Here, we treat a piece of information as our basic unit and use them for further selection.

2. **Check for Key Information in Responses:** For each piece of key information in the reference, we determine whether it is included in each of the candidate answers. Since key information may consist of a main argument accompanied by detailed explanations, we focus only on whether the main argument is present in the answers, disregarding the detailed descriptions. If an answer includes the main argument but provides its own detailed explanation, it should still be considered correct.

3. **Handling Responses:** For similar statement pairs in reference and response, we categorize them into three groups, i.e., **Totally Correct**, **Partially Correct**, and **Totally Wrong**.

   - **Totally Correct:** If an answer contains information that is completely accurate and directly relevant to the question, it is considered totally correct. Such answers contribute fully to the overlapping information score.
   - **Partially Correct:** If an answer contains information that is only partially accurate or relevant, it is considered partially correct. The impact of this information on the overlapping score is reduced, depending on the extent of correctness and relevance.
   - **Totally Wrong:** If an answer contains information that is entirely incorrect or irrelevant to the question, it is considered totally wrong. Such answers should have their overlapping information score set to 0.

4. **Compare Overlapping Information:** The final comparison between the two answers will be based on the amount of overlapping key information. For example, if the reference contains three key pieces of information, and Answer A contains two of them while Answer B contains one, Answer A will be considered the better response.

We show our annotation example in Figure 5.

### B.2  REFERENCE TRANSFORMATION ANNOTATION

To demonstrate the effectiveness and accuracy of our transformation of references into more natural language, we randomly sampled 100 examples from our dataset and conducted human annotation based on the following aspects:

| Type | Geography | History | Politics | Psychology | Medicine | Law |
|---|---|---|---|---|---|---|
| Factuality Error | 0 | 0 | 0 | 0 | 0 | 0 |
| Newly Added Information | 1 | 0 | 0 | 0 | 0 | 0 |
| Error Expansion | 0 | 0 | 0 | 0 | 0 | 0 |

Table 6: Numbers of errors occur in sampled examples. We sample 100 examples from Geography, History, Politics, Psychology, Medicine, and Law and human annotate errors occurred in the transformed references compared with original references.

1. **Factuality Error:** Whether there is a factual error in the transformed references. Annotators are required to verify the accuracy by cross-referencing with the corresponding Wikipedia entries.

2. **Newly Added Information:** Whether there is newly added information in the transformed references. This is assessed by comparing the number of key information points in the original reference to those in the transformed reference. Note that each key information point may include a main argument followed by detailed explanations. The expansion of explanations is not considered newly added information.

3. **Improper Expansion:** Whether the expansion of explanations is improper. The transformation primarily involves expanding the detailed explanations of each main argument. Annotators are asked to evaluate the fluency and topic-centric nature of these expansions.

The results are shown in Table 6 . This indicates that prompting GPT-4o to expand the original references results in high fluency yet knowledge-equivalent references, ensuring the accuracy of our transformation process.

### B.3    CASE STUDY

**Case Study on Reference Transformation:**    An example of reference transformation is presented in Figure 6.

**Case Study on Justification for Disagreement:**    An example of justification for inter-annotator disagreement, resolved by a meta-reviewer, is shown in Figure 7.

**Case Study on Totally Correct, Partially Correct, and Totally Wrong Responses:**    An example illustrating Totally Correct, Partially Correct, and Totally Wrong responses, based on a reference, is provided in Table 14.

**Case Study on Annotation Process:**    An example of the human annotation process is displayed in Figure 8 to aid understanding.

## C    EXPERIMENTS

### C.1    DESCRIPTION FOR TRAINED METRICS

**Auto-J** Auto-J is a model trained using both pairwise data and single-response data from the dataset provided by Zheng et al. (2023). It is capable of transferring to 58 real-world scenarios and can provide detailed natural language explanations for its evaluations.

**TIGERScore** TIGERScore is a model trained on the MetricInstruct dataset, where it is provided with a context and a hypothesis to output an error analysis. Based on Llama-2-13B, it tests hypotheses that may contain errors, listing the identified errors and providing a final score as its rating.

**CritiqueLLM** CritiqueLLM is trained on the Eval-Instruct dataset, which includes pairwise comparisons and pointwise grading. This model demonstrates performance comparable to GPT-4o and can also generate natural language explanations for its evaluations.

## C.2 Main Experiments Without Tie Option

We present the results after excluding the records human annotated as a tie in Table 15. This transforms the evaluation into an A/B choice. The performance does not show overall significant improvement compared to the triple setting mentioned earlier, indicating that automatic evaluation metrics struggle to determine which answer has more semantic overlap with a given reference. This further underscores the challenges inherent in LFQA evaluation. Additionally, GPT-4o with Chain-of-Thought (CoT) still outperforms all other evaluation metrics, demonstrating the superiority of GPT-4o and the effectiveness of the CoT approach.

## C.3 Detailed Correlation between Metrics

We additionally present the detailed correlations for Geography, History, and Politics under the three settings in Figure 9, Figure 10, and Figure 11. The experimental results suggest that when evaluating two responses of similar lengths, LLM-based evaluation metrics do not significantly suffer from the "always long answer" dilemma. However, in the *Human vs. Model* setting, where model-generated responses are typically longer than human-written ones, these metrics tend to favor the longer response, even if it does not contain more information that overlaps with the reference.

## C.4 Do Automatic Evaluation Metrics Mirror Human Challenges?

We present the results in Table 16 and Table 17, illustrating two scenarios: one where there is a disagreement between annotators but the metrics align with the meta-reviews and another where both annotators prefer the same answer and the metrics concur with their decision. Intuitively, when humans struggle to compare two answers, LLMs may also face similar difficulties, resulting in lower agreement rates. Upon examining the results from both tables, we observe that, on average, performance in the first scenario is lower than in the second across most metrics, though the difference is moderate. However, a closer analysis at the micro level reveals that for certain subjects, performance in the second scenario surpasses that of the first, suggesting subject-specific differences and more complex challenges when applying automatic evaluation metrics for LFQA evaluation. These findings indicate that while automatic evaluation metrics may share some challenges with human annotators, they also encounter additional, distinct difficulties. A more detailed analysis can be conducted by breaking down the evaluation process into discrete steps to gain deeper insights into each phase.

## C.5 Error Analysis

Figure 12 illustrates why ROUGE fails when evaluating LFQA. The fundamental issue stems from the inherent logic of ROUGE, which calculates similarity based on word overlap. In the context of LFQA, where answers can be quite lengthy, keywords may appear sporadically, leading to an inaccurate assessment of the overall quality and relevance of the response.

Figure 13 illustrates why LLMs struggle with evaluating LFQA. The primary issue is that LLMs often fail to focus on the key information related to the given question, leading to scores that are not strictly based on the critical information from the reference.

## C.6 Agents Implementation and Details

**CHATEVAL:** We implement CHATEVAL using three agents in one round one-to-one discussion. The three agents are Criric, General Public, and Engineer. They are based on GPT-3.5-turbo, GPT-3.5-turbo-1106-preview, and GPT-3.5-turbo-0613-preview with a temperature equal to 0.7. The role descriptions are listed in Figure 14.

**REFER:** We implement REFER using three reviewers—GPT-3.5-turbo, GPT-4o, and GPT-3.5-turbo-1106-preview—and one area chair role, also played by GPT-4o. All temperatures are set to 0.7. Although the original methodology used models from different organizations, Chan et al. (2023) emphasizes that the role-playing aspect is more critical than the specific models used, so no significant degradation in performance is expected when using models exclusively from OpenAI.

---

*Context:*

The taxation system of the Tang Dynasty was characterized by the principle that "those who own land pay rent, those who have households pay taxes, and those who are able-bodied provide labor." Under this system, members of the royal family, officials, aristocrats, filial sons, loyal wives, and wandering or displaced people would seek protection from local landlords and powerful figures as dependents to avoid paying taxes and performing labor services required by the state. Consequently, only 30% to 50% of households and 14% of the population shouldered the entire tax burden of the state, leading to an extremely inequitable tax system. In the first year of Jianzhong (780 AD), Emperor Dezong issued an order: "Let the Commissioners for Examinations and Promotions, along with the county officials, assess the old tax revenues and the conditions of households, both native and non-native, to determine the amount of money to be collected for the summer and autumn taxes." The two-tax system implemented a "budgeting based on expenditures" principle. Each year, the government would first calculate the total fiscal expenditure and then allocate it according to the actual conditions of land, population, and other factors in each region, ensuring a difference in tax burdens between the rich and poor while maintaining equitable taxation. This principle prevented local officials from extorting the people, collecting excessive taxes, or increasing the burden on the populace.

*Question:*

Based on Material Two and combined with the knowledge learned, identify the background of the implementation of the Two-Tax System during the Tang Dynasty and explain the changes in taxation that it caused.

*Reference:*

The background includes widespread land annexation, the destruction of the Equal-field system, the inability to implement the rent, labor, and tax system, and a decrease in national fiscal revenue. The changes were that the tax standards were primarily based on assets, the tax collection time became fixed, tax categories were simplified, the number of taxpayers increased, and the authority to assess taxes was further centralized under the imperial court.

---

Table 7: An example of a Factoid Question including context, question, and reference.

During the evaluation, we prompt the LLM agents to focus on information that overlaps with the reference. We first obtain ratings for two answers and then compare them to make the final decision.

C.7   PROMPTS

**Prompt for Reference Transformation:**   The prompt used for Reference Transformation is provided in Table 18.

**Prompt for ChatGPT:**   The prompt used for ChatGPT is shown in Table 19.

**Prompt for G-Eval:**   The prompt used for G-Eval is displayed in Table 20.

**Prompt for CHATEVAL:**   The prompt used for CHATEVAL is presented in Table 21.

**Prompt for REFER:**   The prompts used for REFER, including the prompts for reviewers, are provided in Table 22, and the prompt for the area chair (AC) is provided in Table 23.

*Context:*

After the War of Independence, the original Confederation government became nominal and on the verge of collapse as each state sought to protect the interests of its ruling elite. At the same time, foreign powers, led by Britain, were resentful of and hostile toward the United States' independence. The founders of American constitutionalism began to further systematize and Americanize Montesquieu's theories, establishing the principle of separation of powers and checks and balances. When the Constitutional Convention was convened in 1787, there was no consensus among the affluent bourgeoisie. However, when Shays' Rebellion of 1786-1787 threatened their fundamental interests, these differences were replaced by consensus. The U.S. Constitution grants legislative power to Congress, free from interference by the executive branch, and gives Congress the power to impeach the president. The executive power is exercised by the president, who is elected by the voters and is accountable only to them. The president has the power to veto legislation passed by Congress. Judicial power is vested in the Federal Supreme Court and any lower courts that Congress may establish at any time. The Supreme Court has the power of final adjudication, and federal judges are appointed by the president with the consent of the Senate. Judges may serve for life as long as they faithfully perform their duties and cannot be removed except by impeachment by Congress. Additionally, the Constitution stipulates that no official of one branch of government may hold a position in another branch during their term of office. Excerpted and adapted from Wei Haiqun's "The Historical Development and Implications of the U.S. Separation of Powers System."

*Question:*

Based on Material Two and combined with the knowledge learned, summarize the reasons for the establishment of the separation of powers system in the United States. Explain the main differences between the power separation system in the Tang Dynasty and the separation of powers system in the United States.

*Reference:*

The reasons include the victory of the American War of Independence, the influence of Enlightenment thought, the impetus from farmers' uprisings, and the need for practical political development. The main differences are that the separation of powers in the Tang Dynasty was based on the historical tradition of dividing the powers of the chancellors from previous dynasties, while the separation of powers in the United States was based on a pre-existing democratic tradition before the founding of the nation. The Tang Dynasty's separation of powers primarily involved the parallel division of drafting, issuing, and executing powers, whereas the American separation of powers involved the intersecting division of legislative, executive, and judicial powers. The Tang Dynasty's separation of powers aimed to strengthen the autocratic rule of the emperor, while the American separation of powers was designed to prevent despotism and tyranny.

Table 8: An example of a Comparative Question including context, question, and reference.

*Context:*

*(No context provided in the original document)*

*Question:*

Briefly describe the types of criminal objects.

*Reference:*

The object of a crime refers to the social interests that are harmed by criminal activities and are protected by criminal law. The object of a crime can be classified into three types based on the scope of the social relationships being harmed: general object, similar object, and direct object. The general object refers to the social interests that are harmed by all crimes collectively, which is the overall social interest. Both the direct object and the similar object are components of the general object of social interests, and the three have a relationship of particular, partial, and whole. The similar object refers to the social interests that are harmed by a certain category of crimes. The similar object of a crime encompasses the common characteristics of a category of crimes and serves as the basis for classifying crimes. The direct object refers to the specific social interests directly harmed by a particular crime. The direct object of a crime is a component of the constitution of a particular crime, directly reflecting the social nature of the interests harmed by that criminal behavior. The direct object can be further divided into simple object and complex object.

Table 9: An example of a Definition Question including context, question, and reference.

*Context:*

Corn oil is a type of grain oil produced from corn germ, known for its rich nutritional content and pleasant flavor. The production process of corn oil consists of crude oil extraction and refining stages. The oil extraction rate from germ is 40%, and the conversion rate from crude oil to refined oil is 90%. A company in Zouping, Shandong, is the earliest established and currently the largest corn oil product research and production enterprise in China, with its corn oil sales accounting for 50% of the domestic market. The company has crude oil pressing plants in Huimin, Shandong; Tieling, Liaoning; Tongliao and Ordos, Inner Mongolia. It also has refining oil and small packaging product production bases at its headquarters in Hangzhou, Zhejiang; Guangzhou, Guangdong. The crude oil is often transported to the refining oil production bases using flexitanks (disposable soft packaging containers used for storing and transporting various non-hazardous liquid goods), while the use of tank containers or steel drums for transportation is becoming increasingly rare.

*Question:*

Analyze the main reasons for the company's establishment of refining oil and small packaging product production bases in Hangzhou and Guangzhou.

*Reference:*

Proximity to the market, large population coverage, and substantial market size cater to local consumer demands. The cost of transporting crude oil is similar to that of transporting refined oil, and the process of transitioning from refined oil production to small packaging is seamless, allowing products to be quickly introduced to the market, which helps ensure product freshness.

Table 10: An example of an Analytical Question including context, question, and reference.

*Context:*

Grain production is influenced by various factors, including the amount of agricultural fertilizer used, population size, regional economic development level, mechanization level, natural disasters, technology, and agricultural support policies. The Yangtze River Economic Belt has a solid economic foundation and favorable agricultural production conditions, making it a traditional major grain-producing region and a key commodity grain base in China. As an important grain-producing area in the south, the grain output of the Yangtze River Economic Belt has generally shown an upward trend. However, its share of the national grain output has been declining, from 49.4% in 1978 to 36.35% in 2018. The share of agricultural output value in the national economy has also decreased, from 17% to 6.7%.

*Question:*

Some people believe that the improvement in economic levels in the Yangtze River Economic Belt is detrimental to the development of grain security production. Do you agree? State your opinion and provide reasons.

*Reference:*

I agree, and the reasons are that with economic development, the rapid advancement of industrialization and urbanization will inevitably encroach on agricultural land, leading to a reduction in the area of grain cultivation. The development of industrialization may exacerbate environmental pollution, affecting grain quality. Economic development drives the migration of surplus rural labor to cities, resulting in a shortage of young and able-bodied workers in rural areas, which impacts grain production. Alternatively, one might disagree, arguing that economic development allows the government to increase financial support for agriculture, enhance the construction of water conservancy facilities, boost farmers' enthusiasm for grain cultivation, and improve agricultural production conditions and management levels. Economic development facilitates farmers' increased investment in agricultural technology and production materials, which can improve grain yield and quality. Economic development can also enhance the operational levels of grain processing enterprises, further promoting grain production.

Table 11: An example of an Evaluation Question including context, question, and reference.

*Context:*

In recent years, H City has been enhancing the well-being of its citizens by creating a better ecological and green environment through its Park City construction project. Residents can reach a park within a 10-minute walk or a 5-minute bike ride. Park construction serves as a public welfare initiative, ensuring that the benefits of development are more equitably shared among all people. Moreover, by integrating the "Park+" and "+Park" concepts, it promotes the innovation of urban development ideas, facilitating the transformation from "city parks" to a "Park City." This transformation has elevated the city's status, making H City more attractive to new talent and industries, stimulating tourism development, and positively impacting local employment.

*Question:*

Using knowledge from "Economy and Society," explain how H City has implemented the new development concepts in the process of building the "Park City."

*Reference:*

Adhering to green development: H City has created a shared ecological and green environment, promoting harmony between people and nature. Adhering to shared development: The city ensures that the benefits of development are more equitably shared among all people, meeting their pursuit of a better life. Adhering to innovative development: The city fosters innovation in urban development concepts through the "Park+" and "+Park" approaches. Adhering to coordinated development: By enhancing the city's status, H City stimulates economic growth and promotes the coordinated development of the economy and society.

Table 12: An example of an Inferential Question including context, question, and reference.

*Context:*

In recent years, driven by emerging technologies such as the Internet, big data, and artificial intelligence, China's digital economy has developed rapidly. The digital economy, centered on digital technology innovation, with data resources as key elements and modern information networks as important carriers, has provided new and powerful momentum for economic and social development. However, the rapid development of digital technology has also brought a series of challenges, such as data security, personal information protection, and network regulation. To promote the healthy development of the digital economy in China, the government has taken a series of measures, including strengthening digital infrastructure construction, optimizing the digital business environment, and improving the legal and regulatory framework for the digital economy. The implementation of these measures has provided strong support for the development of the digital economy.

*Question:*

Based on the material, discuss the measures the government should take to improve the legal and regulatory framework for the digital economy.

*Reference:*

Formulate and improve laws and regulations related to the digital economy to provide legal protection for its development. Strengthen the regulation of the digital economy to ensure fair market competition and protect consumer rights. Establish and improve mechanisms for data security and personal information protection to safeguard data security and individual privacy rights. Promote the coordinated development of the digital economy with other sectors, achieving deep integration between the digital economy and other industries.

Table 13: An example of an Application Question including context, question, and reference.

*Reference:*

The objective aspects are specifically manifested in any of the following behaviors: randomly beating others, in a particularly egregious manner; chasing, intercepting, insulting, or intimidating others, in a particularly egregious manner; forcibly taking or deliberately destroying or occupying public or private property, in a particularly serious manner; or causing serious disorder in a public place by inciting trouble.

*Totally Correct Answer:*

The objective aspects include behaviors such as randomly beating others in a particularly egregious manner, chasing or intimidating others in a particularly egregious manner, forcibly taking or destroying public or private property in a particularly serious manner, and causing serious disorder in public places by inciting trouble.

*Partially Correct Answer:*

The objective aspects involve behaviors like beating others or intimidating them, forcibly taking or occupying property, and causing disorder in public places. However, it lacks the emphasis on the seriousness and egregious nature of these actions as described in the original text.

*Totally Wrong Answer:*

The objective aspects include behaviors like helping others in public, being polite to strangers, and following the rules in public places. This answer is completely incorrect as it misrepresents the negative behaviors described in the original text as positive actions.

Table 14: An example showing Totally Correct Answer, Partially Correct Answer, and Totally Wrong Answer.

Question:

What is an instigator in Chinese criminal law?

**1. Identify Key Information in reference**

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

Which will be considered the better response?

Answer A          Answer B

Figure 5: The overall pipeline of our annotation process. It can be separated into four stages, i.e., **Identify Key Information**, **Handling Responses** and **Check for Key Information in Answers**, **Compare Overlapping Information**. The texts denoted by blue are key information identified in reference and answers.

**From Geo 335**:
Briefly describe the advantageous conditions that attract the company to build crude oil pressing factories in Huimin, Tieling, Tongliao, and Ordos.

**Key information:**
- These areas are major corn-producing regions with abundant raw materials;
- They are underdeveloped small and medium-sized cities with cheap land prices and abundant, low-cost labor.

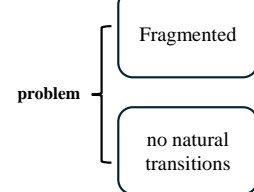

problem → Fragmented / no natural transitions

**Reference:**
Let's discuss this issue together. These factors collectively drive the development of the region. **Firstly**, these areas are major corn-producing regions with abundant raw materials, ensuring a sufficient supply of raw materials for agricultural product processing and related industries. **Secondly**, these regions are underdeveloped small and medium-sized cities with cheap land prices, reducing operational costs for enterprises. **Additionally**, abundant and low-cost labor provides companies with a large amount of inexpensive labor, further reducing production costs. These advantages make the region well-suited for the development of agricultural product processing and related industries.

Figure 6: An example showing the reference transformation results. The sentences denoted in red and green contain the key information used to answer the questions.

## Justification 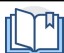

Context:
The Guizhou Plateau is a karst plateau with a highly fragile ecological environment. The diurnal variations in water and heat conditions and their impact on vegetation changes can provide theoretical support for ecological protection and disaster prevention and mitigation in this region.

Question:
Explain the reasons why the ecological environment in Guizhou Province is highly fragile.

Reference:
Guizhou Province has rugged terrain, primarily characterized by karst topography. The landforms develop slowly, with thin soil layers. Abundant rainfall leads to strong water erosion, making the soil prone to loss. The underground is filled with caves and fissures, causing significant infiltration of surface water, which results in surface water scarcity.

Answer_a:
There are four main reasons why the ecological environment in Guizhou Province is highly fragile:Firstly, Guizhou has a karst topography with undulating terrain and complex landforms.Secondly, severe soil erosion has led to significant depletion of land resources.Thirdly, plains are scarce in Guizhou, with most areas being mountainous, making them more susceptible to natural disasters.Fourthly, the rugged terrain and difficult transportation conditions pose challenges to ecological protection and restoration efforts.

Answer_b:
There are three main reasons why the ecological environment in Guizhou Province is highly fragile: Firstly, the unique karst topography is easily affected by water erosion and dissolution, leading to the formation of distinctive landforms; Secondly, the humid climate with abundant rainfall intensifies the erosion processes; Lastly, the exceptionally high nighttime rainfall, coupled with insufficient vegetation cover, results in severe soil erosion.

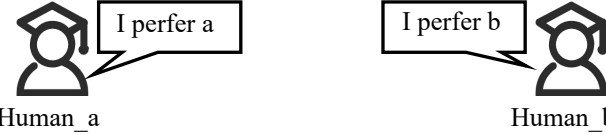

Analysis:
Human_A leans more towards Answer A because it mentions severe soil erosion, while Answer B only refers to erosion without directly stating severe soil erosion. On the other hand, Human_B prefers Answer B because it mentions abundant rainfall, which Answer A does not mention. However, erosion and soil erosion share the same implied meaning, and Answer A completely omits the important point about rainfall. Therefore, ultimately preferring Answer B is more reasonable.

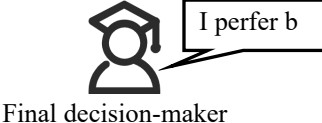

Final decision-maker

Figure 7: An example showing the deal with disagreement between annotators. When a disagreement appears, a final annotator will make the decision and give the justification.

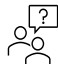

**From Geo 79**:
Briefly describe the reasons for the decrease in oyster size in recent years

**Reference:**
With the rapid development of the oyster farming industry, the number of oysters has increased dramatically, leading to reduced living space and limited food sources. Additionally, the construction of reservoirs upstream and the nighttime release of water have increased the freshwater content in the river, affecting the oysters' growth environment.", "Reference": "I'm happy to answer your question. With the rapid development of the oyster farming industry, the number of oysters has increased dramatically, which is a positive development but also brings some issues. First, the living space for oysters has become increasingly crowded. Second, food sources have become quite limited, leading to competition among oysters. Additionally, the construction of reservoirs upstream, intended to better manage water resources, has resulted in increased freshwater content in the river due to nighttime releases. This change has also impacted the growth environment for oysters. In summary, although the oyster farming industry is growing rapidly, these potential issues need attention to ensure sustainable development.

**Answer a:**
With the rapid development of the oyster farming industry, the number of oysters has increased dramatically. As a result, their living space has shrunk, and food sources have become more limited. Additionally, areas near the river provide necessary production and living water for industry, which has impacted oyster growth.

**Answer b:**
Due to the rapid increase in the number of oysters, their living space has decreased, and food sources have also diminished. However, in water-rich regions like the Yangtze and Yellow Rivers, fisheries and hydropower industries can still be developed.

**Analysis:**
**Key Points from the Standard Answer:**
1. Rapid development of the oyster farming industry: The industry has grown quickly, leading to a significant increase in the number of oysters.
2. Reduced living space: The increased number of oysters has led to overcrowding and reduced space for their growth.
3. Limited food sources: The overcrowding has also resulted in limited food availability, causing competition among oysters.
4. Impact of upstream reservoir construction: The construction of reservoirs upstream has increased freshwater content in the river.
5. Freshwater content affecting growth environment: The nighttime release of water has altered the river's composition, impacting the oysters' growth environment.

**Answer A** mentions the key points about the increase in oysters, reduced living space, and limited food sources, aligning well with the standard answer. However, it briefly touches on the environmental impact without detailing the freshwater changes.
**Answer B** discusses the reduction in space and food but overlooks the crucial impact of reservoir construction and freshwater changes on oyster growth, making it less aligned with the standard answer.
**Therefore, Answer A** is closer to the requirements of the standard answer and is more comprehensive.

Figure 8: An example showing the human annotation process,

| Metric | Geography | History | Politics | Psychology | Medicine | Law | Avg. |
|---|---|---|---|---|---|---|---|
| Traditional Metrics | | | | | | | |
| ROUGE | 0.541 | 0.513 | 0.454 | 0.547 | **0.747** | 0.638 | 0.573 |
| BLEU | 0.350 | 0.372 | 0.411 | 0.520 | 0.627 | 0.617 | 0.483 |
| BERTScore | 0.474 | 0.421 | 0.379 | 0.413 | 0.542 | 0.383 | 0.435 |
| BLEURT | 0.574 | 0.474 | 0.540 | 0.520 | 0.518 | 0.489 | 0.519 |
| BARTScore | 0.562 | 0.508 | 0.563 | 0.613 | 0.723 | 0.585 | 0.592 |
| UniEval | 0.612 | 0.503 | 0.431 | 0.373 | 0.675 | 0.543 | 0.523 |
| GPT2♠ | 0.494 | 0.490 | 0.532 | 0.547 | 0.735 | 0.596 | 0.566 |
| Prompt-only Metrics | | | | | | | |
| ChatGPT (0613) | **0.588** | 0.533 | 0.563 | 0.680 | 0.711 | 0.543 | 0.603 |
| ChatGPT-CoT | 0.574 | 0.518 | **0.601** | 0.653 | 0.735 | 0.553 | 0.606 |
| G-Eval (ChatGPT) | 0.582 | 0.526 | 0.592 | 0.667 | 0.711 | **0.638** | 0.619 |
| GPT-4o | 0.541 | 0.513 | 0.586 | 0.653 | 0.687 | 0.628 | 0.601 |
| GPT-4o-CoT | 0.553 | **0.561** | 0.589 | **0.693** | 0.699 | **0.638** | **0.622** |
| G-Eval (GPT-4o) | 0.526 | 0.548 | 0.592 | 0.680 | 0.735 | 0.511 | 0.599 |
| Trained Metrics | | | | | | | |
| Critique-6B | 0.459 | 0.528 | 0.483 | 0.427 | 0.410 | 0.351 | 0.443 |
| AutoJ-13B | 0.494 | 0.508 | 0.546 | 0.627 | 0.699 | 0.628 | 0.584 |
| TIGERScore-13B♠ | 0.415 | 0.268 | 0.115 | 0.120 | 0.265 | 0.160 | 0.224 |

Table 15: Evaluation Metrics Across Different Domains. The baselines denoted by ♠ are reference-free evaluation metrics. The matching rate of model preference and human preference in the triple-choice settings measures all the results. The largest accuracy rate is denoted using **bold**.

| Metric | Geography | History | Politics | Psychology | Medicine | Law | Avg. |
|---|---|---|---|---|---|---|---|
| Traditional Metrics | | | | | | | |
| ROUGE | 0.500 | 0.466 | 0.551 | 0.392 | 0.333 | 0.541 | 0.464 |
| BLEU | 0.538 | 0.427 | 0.482 | 0.307 | 0.333 | 0.500 | 0.431 |
| BERTScore | 0.423 | 0.437 | 0.517 | 0.331 | 0.389 | 0.375 | 0.412 |
| BLEURT | 0.307 | 0.427 | 0.276 | 0.484 | 0.489 | 0.416 | 0.400 |
| BARTScore | **0.615** | **0.475** | 0.482 | 0.380 | 0.500 | 0.583 | 0.506 |
| UniEval | 0.461 | **0.475** | 0.586 | 0.503 | 0.400 | 0.291 | 0.453 |
| GPT2♠ | 0.384 | 0.407 | 0.551 | 0.423 | 0.533 | 0.500 | 0.466 |
| Prompt-only Metrics | | | | | | | |
| ChatGPT | 0.500 | **0.475** | 0.517 | 0.466 | 0.489 | 0.750 | 0.533 |
| ChatGPT-CoT | 0.500 | 0.427 | 0.517 | 0.460 | **0.544** | 0.708 | 0.526 |
| G-Eval (ChatGPT) | 0.613 | 0.446 | 0.482 | 0.429 | **0.544** | 0.708 | 0.537 |
| G-Eval (ChatGPT) | 0.613 | 0.446 | 0.482 | 0.429 | **0.544** | 0.708 | 0.537 |
| GPT-4o | 0.538 | 0.446 | 0.482 | 0.398 | 0.500 | **0.791** | 0.526 |
| GPT-4o-CoT | 0.500 | 0.466 | **0.655** | 0.466 | 0.500 | 0.666 | **0.542** |
| G-Eval (GPT-4o) | 0.500 | 0.466 | 0.586 | 0.435 | 0.489 | 0.666 | 0.524 |
| Trained Metrics | | | | | | | |
| Critique-6B | 0.230 | **0.475** | 0.276 | **0.582** | 0.366 | 0.375 | 0.384 |
| AutoJ-13B | 0.384 | 0.437 | 0.448 | 0.392 | 0.500 | 0.666 | 0.471 |
| TIGERScore-13B♠ | 0.192 | 0.395 | 0.241 | 0.233 | 0.189 | 0.041 | 0.215 |

Table 16: The agreement rate of the situation where there is a disagreement between annotators but the metrics make the same decision as the meta-reviewer. The baselines denoted by ♠ are reference-free evaluation metrics. ChatGPT used here is the GPT-3.5-turbo-1106-preview version. The largest agreement rate is denoted using **bold**.

.

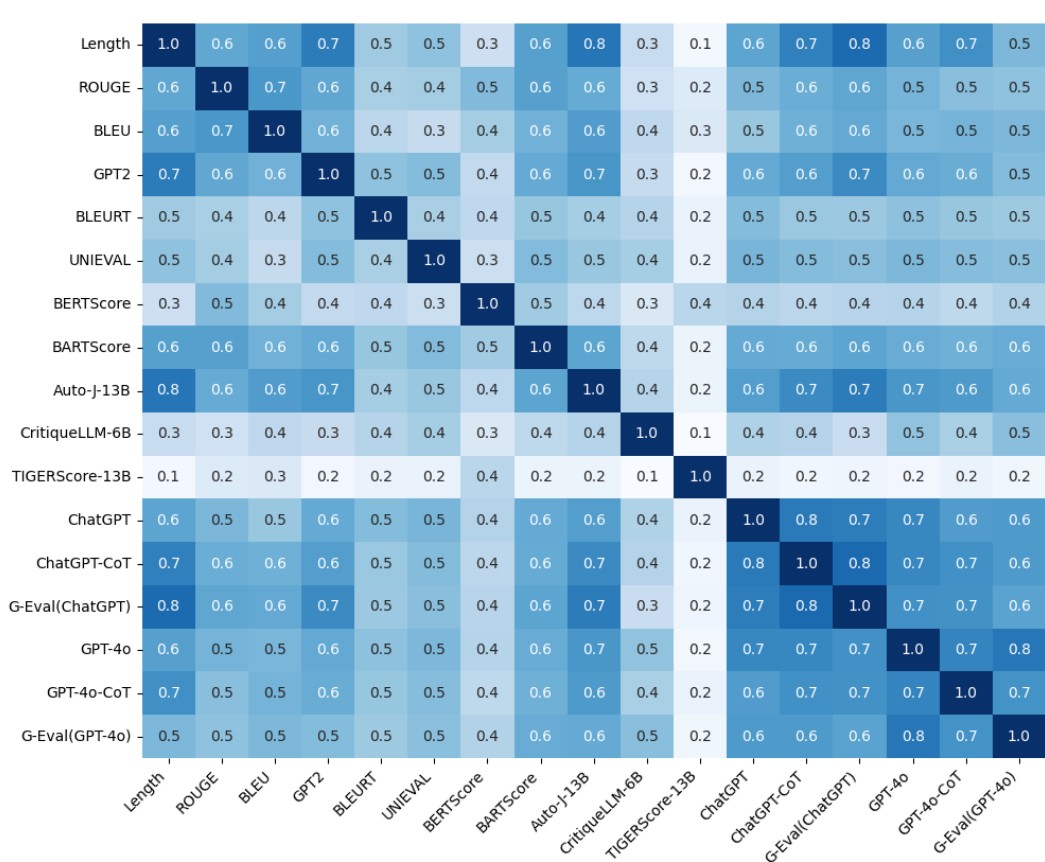

Figure 9: Overall correlation between automatic evaluation baselines on *Huamn vs. Human*. The lighter color indicates a little correlation while the darker color indicates a larger correlation. The annotation on each block is calculated by the ratio of the intersection between $BaselineA$ and $BaselineB$ and the total number of records tested by both $BaselineA$ and $BaselineB$.

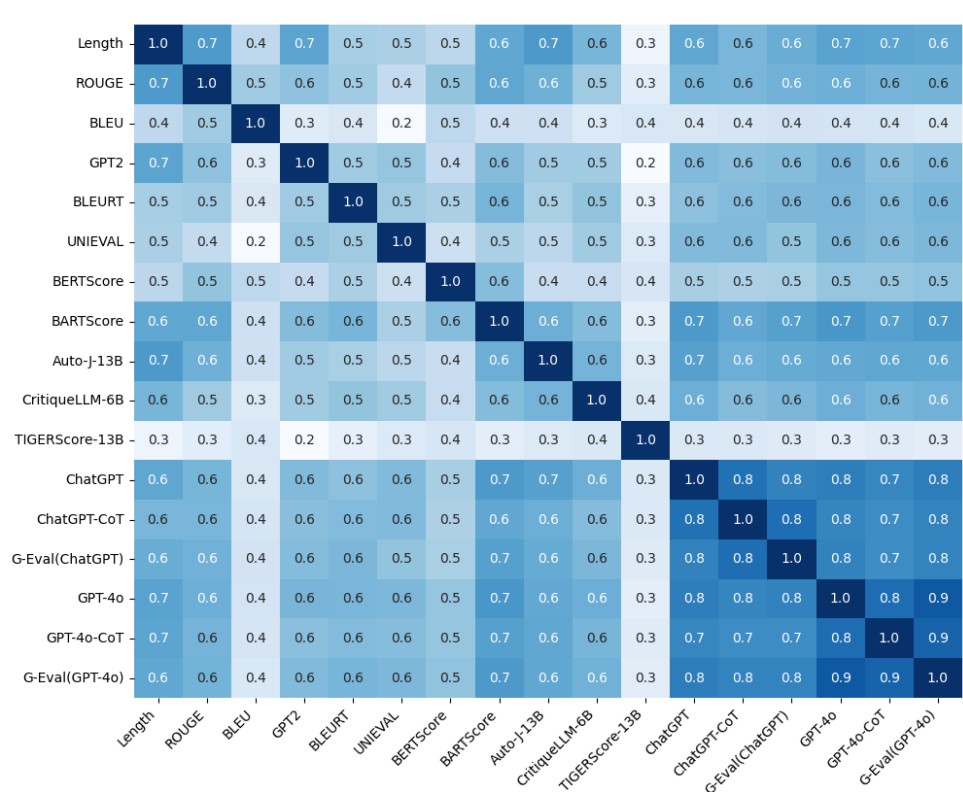

Figure 10: Overall correlation between automatic evaluation baselines on *Human vs. Model*. The lighter color indicates a little correlation while the darker color indicates a larger correlation. The annotation on each block is calculated by the ratio of the intersection between $BaselineA$ and $BaselineB$ and the total number of records tested by both $BaselineA$ and $BaselineB$.

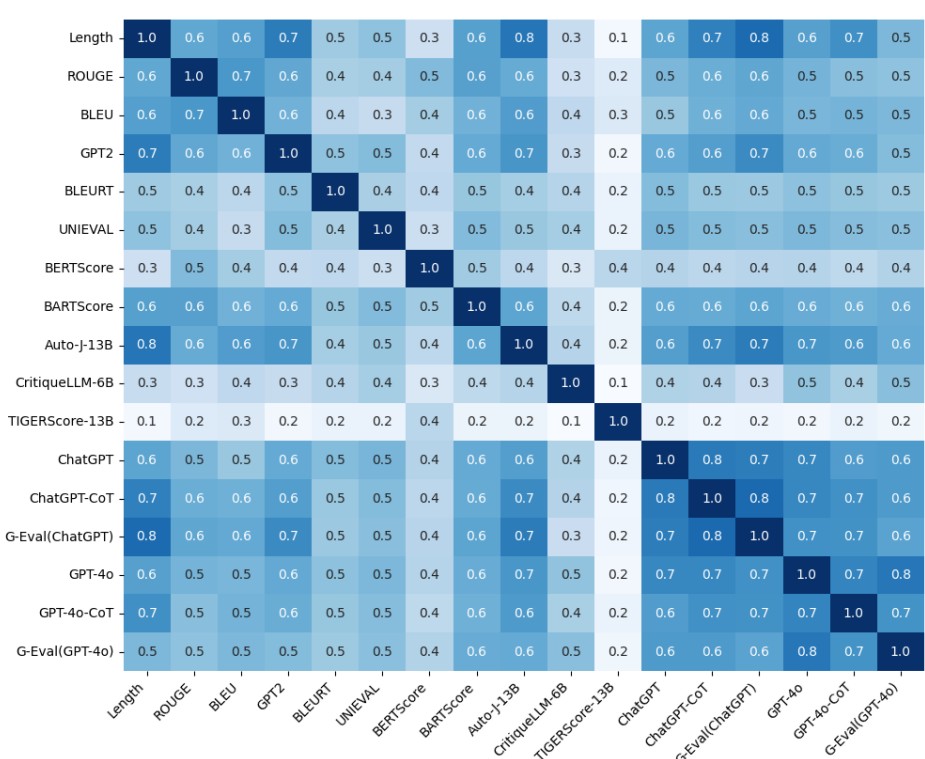

Figure 11: Overall correlation between automatic evaluation baselines on *Model vs. Model*. The lighter color indicates a little correlation while the darker color indicates a larger correlation. The annotation on each block is calculated by the ratio of the intersection between $Baseline A$ and $Baseline B$ and the total number of records tested by both $Baseline A$ and $Baseline B$.

| Metric | Geography | History | Politics | Psychology | Medicine | Law | Avg. |
|---|---|---|---|---|---|---|---|
| Traditional Metrics | | | | | | | |
| ROUGE | 0.457 | 0.598 | 0.441 | 0.417 | **0.657** | 0.635 | 0.534 |
| BLEU | 0.370 | 0.419 | 0.402 | 0.472 | 0.557 | 0.622 | 0.474 |
| BERTScore | 0.426 | 0.485 | 0.376 | 0.458 | 0.514 | 0.378 | 0.440 |
| BLEURT | 0.495 | 0.467 | 0.477 | 0.431 | 0.500 | 0.514 | 0.481 |
| BARTScore | 0.495 | 0.598 | 0.506 | 0.444 | 0.657 | 0.527 | 0.538 |
| UniEval | 0.481 | 0.502 | 0.376 | 0.333 | 0.557 | 0.527 | 0.463 |
| GPT2♠ | 0.436 | 0.537 | 0.448 | 0.403 | 0.629 | 0.622 | 0.512 |
| Prompt-only Metrics | | | | | | | |
| ChatGPT | 0.484 | 0.581 | 0.497 | 0.458 | 0.629 | 0.527 | 0.529 |
| ChatGPT-CoT | **0.505** | 0.559 | 0.526 | 0.458 | 0.643 | 0.527 | 0.536 |
| G-Eval (ChatGPT) | 0.502 | 0.594 | 0.513 | 0.458 | **0.657** | 0.608 | 0.555 |
| GPT-4o | 0.495 | 0.594 | 0.520 | 0.417 | 0.614 | 0.608 | 0.541 |
| GPT-4o-CoT | 0.498 | **0.629** | 0.529 | **0.528** | 0.600 | **0.662** | **0.574** |
| G-Eval (GPT-4o) | 0.481 | **0.629** | **0.533** | 0.486 | 0.643 | 0.473 | 0.541 |
| Trained Metrics | | | | | | | |
| Critique-6B | 0.453 | 0.489 | 0.441 | 0.319 | 0.371 | 0.365 | 0.407 |
| AutoJ-13B | 0.457 | 0.590 | 0.474 | 0.444 | 0.643 | 0.662 | 0.545 |
| TIGERScore-13B♠ | 0.374 | 0.293 | 0.219 | 0.361 | 0.357 | 0.176 | 0.297 |

Table 17: The agreement rate of the situation where both annotators prefer one answer and the metrics make the correct decision. The baselines denoted by ♠ are reference-free evaluation metrics. ChatGPT used here is the GPT-3.5-turbo-1106-preview version. The largest agreement rate is denoted using **bold**.

---

***Instruction:***

You will be provided with a text that is concise, unpolished, and heavily focused on knowledge content. Your task is to transform this text into fluent and natural language without adding any new information.
To achieve this, you can utilize the following strategies:
Expand with Relevant Examples: Use examples where appropriate to clarify or elaborate on the content, making it more engaging and accessible.
Include Auxiliary Information: Add supportive information, such as a brief summary, a self-introduction, or other relevant context, to enhance readability and coherence.
Incorporate Transitional Sentences: Use organizational phrases like "After that..." or "Following this..." to improve the flow and structure of the text.
**Text to Transform**:
[Your Text Here]

---

Table 18: Instruction for transformation of references

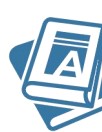

Context:
The Guizhou Plateau is a karst plateau with a very fragile ecological environment. The diurnal thermal conditions and their impact on vegetation changes can provide theoretical support for ecological protection and disaster prevention and reduction in the region.

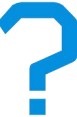

Question:
Analyze the reasons for the frequent nighttime rainfall in Guizhou.

Reference:
The terrain is uneven with many valleys; at night, there are prevailing upward air currents in the valleys, where water vapor condenses easily, leading to precipitation. The area has abundant vegetation, strong evapotranspiration, and the mountainous water vapor does not disperse easily, ensuring sufficient moisture.

Answer_a:
The Karst landscape in Guizhou is characterized by mountains formed by water erosion, strong mountain winds at night, hot regions, and numerous winding rivers. Due to the replacement of water vapor by local hot air, there is less rainfall during the day, resulting in more precipitation at night in the region.

Answer_b:
Large changes in temperature throughout the day, warm air cooling and turning into rain when it meets cold air, affected by almost-stationary fronts, lots of plants and strong evaporation and transpiration.

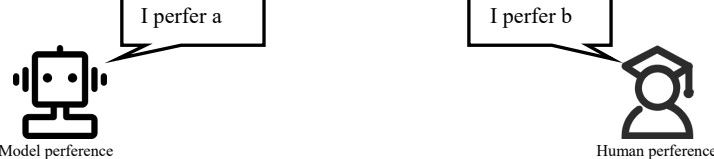

Analysis:
The reason ROUGE scores tend to favor answer A lies mainly in its reliance on word overlap. Answer A uses terms like "mountains," "water vapor," and "precipitation," which closely match the keywords in the reference information, leading to a higher surface similarity score. For example, A mentions "water vapor being replaced by local hot air," which has some lexical connection to the reference information's "water vapor does not disperse easily," even though the meanings may differ. On the other hand, answer B includes descriptions like "large day-night temperature differences" and "strong plant transpiration," which are more accurate but have a lower direct word match with the reference. As a result, ROUGE struggles to accurately assess the quality of content in B. This highlights ROUGE's limitations in capturing deeper semantic and logical relationships, causing it to favor the answer with higher surface word overlap, such as A.

Figure 12: Example and explanation of ROUGE for why ROUGE fails at LFQA evaluation.

Context:
"context": "Sardines thrive in water temperatures ranging from 14 to 20°C. The sardine migration from May to September at the Agulhas Bank fishing grounds attracts many tourists.",

Question:
Question: Explain the formation of the Agulhas Bank fishing grounds. 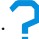

Reference:
The convergence of cold and warm currents causes sea water disturbances, leading to the upwelling of nutrients from the bottom, which promotes the growth of plankton and provides abundant fish food; the convergence of currents creates a water barrier that facilitates fish aggregation; located in the nearshore continental shelf area with freshwater input from land, it is rich in nutrients and plankton; the shallow water allows good light penetration, supporting photosynthesis in plankton.

Student_answer_a:
Ocean currents bring abundant food and create upwellings, contributing to favorable conditions for fish survival in lower latitudes. Additionally, there is ample living space with minimal human impact and pollution.

Student_answer_b:
1. The blend of cold and warm currents helps maintain a stable water temperature. 2. Situated on the continental shelf with shallow waters, allowing for better absorption of light. 3. This area possesses fertile soils and supports prolific plankton growth. 4. The estuarine region is rich in nutrients, providing fish with ample food.",

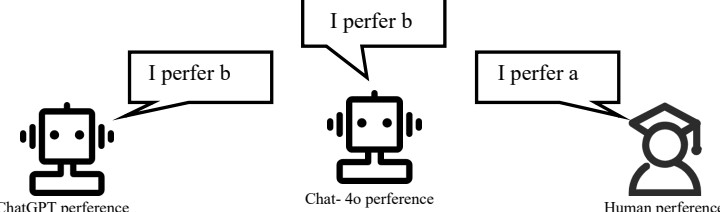

Analysis:
ChatGPT-4o and 3.5 prefer Answer B because it accurately covers all the key points from the reference text, such as the convergence of cold and warm currents, light absorption in shallow waters, and the nutrient-rich estuarine area. Additionally, Answer B presents the information in a clear and structured manner, which aligns with the model's scoring criteria. In contrast, human reviewers are more inclined to prefer Answer A. Although Answer A includes elements like "ample living space" and "minimal human impact and pollution," which are not directly mentioned in the reference, these reasonable inferences and expansions demonstrate deeper thinking. Human evaluators might find Answer A to be concise and reflective of critical reasoning, even though its content diverges somewhat from the core information in the reference text.

Figure 13: Example and explanation of LLMs for why LLMs fail at LFQA evaluation.

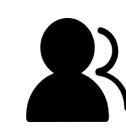

## Critic

You are now Critic, one of the referees in this task. You will check fluent writing, clear sentences, and good wording in summary writing. Your job is to question others judgment to make sure their judgment is well-considered and offer an alternative solution if two responses are at the same level.

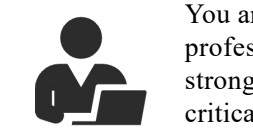

## General Public

You are now General Public, one of the referees in this task. You are interested in the story and looking for updates on the investigation. Please think critically by yourself and note that it's your responsibility to choose one of which is the better first.

## Scientist

You are Scientist, one of the referees in this task. You are a professional engaged in systematic study who possesses a strong backGeneral Publicground in the scientific method, critical thinking, and problem-solving abilities. Please help other people to determine which response is the better one.

Figure 14: Role Descriptions in CHATEVAL.

---

***Instruction:***

Assume you are a teacher. Next, I will provide a paragraph of text containing Reference, Answer A, and Answer B.
You should decide which answer is better based on the Reference.
If you think Answer A is better, please type a. If you think Answer B is better, please type b. Otherwise, please type "tie". You should think step by step before making a decision. Return with "My choice is a" or "My choice is b" or "My choice is tie"."
**Reference:**
[Your Reference Here]
**Answer A:**
[Your Answer A Here]
**Answer B:**
[Your Answer B Here]

---

Table 19: Instruction for ChatGPT to evaluate LFQA

*Instruction:*

**Task Instruction:**

You will be provided with two student responses and a reference answer. Your task is to evaluate and compare the two responses based on the criteria below, and then determine which response is better.

**Evaluation Criteria:**

**Accuracy:**

Compare how closely each response aligns with the content of the reference answer. Consider whether each student correctly identified and conveyed the key points and facts presented in the reference. Identify which response better captures the essential details. **Clarity and Structure:**

Assess the organization, logical flow, and readability of each response. Evaluate the grammar, word choice, punctuation, and sentence structure. Determine which response is clearer and more effectively communicated.

**Completeness:**

Examine whether each response addresses all aspects of the question or key points from the reference answer. Consider the thoroughness of each response, and identify if either response misses or omits important components. Decide which response is more comprehensive.

**Evaluation Process:**

Understand the Reference: Begin by reading the reference answer carefully to grasp the main content, key points, and details.

Analyze Each Response: Read both student responses attentively, ensuring you fully understand what each student is expressing.

Compare Accuracy: Evaluate which response more accurately captures the key points and facts from the reference answer.

Compare Clarity and Structure: Assess which response is better organized, clearer, and more grammatically sound.

Compare Completeness: Determine which response more thoroughly covers all key points and aspects of the reference answer.

**Final Decision:**

After completing your comparison, select the response that best meets the criteria of accuracy, clarity, structure, and completeness. Make your decision by clearly stating which response is better, and provide a brief rationale for your choice.

**Reference**:

[Your Reference Here]

**Answer A**:

[Your Answer A Here]

**Answer B**:

[Your Answer B Here]

Table 20: Instruction for G-Eval

---

***Instruction:***

---

**Context**:
[Your Context Here]
**Question**:
[Your Question Here]
**Reference**:
[Your Reference Here]
**Answer A**:
[Your Answer A Here]
**Answer B**:
[Your Answer B Here]
**Role Description**:
[Your Role Description Here]
**System Message**:
We would like to request your feedback on the performance of two assistants in response to the user question displayed above based on the reference provided. You should decide which one is better based on the reference. You should be critical and your opinion can not be exactly the same as others.
You have received feedback from other referees who have analyzed the responses. Your task is to review this feedback critically and either support or challenge the previous judgments.
Consider if the previous decisions were made correctly. Reflect on whether the initial judgments align with the reference provided. If you disagree with the previous analysis, provide a new perspective and rationale.
Remember, your goal is to ensure the most accurate and fair judgment. You should first analyze the previous feedback and then provide your own feedback.
Here is your discussion history:
[History]
Now it's your time to talk, please make your talk short and clear! You should return with "My decision is a" or "My decision is b" or "My decision is tie" with short and clear explanations.

---

Table 21: Instruction for Agents in CHATEVAL

---

*Instruction:*

---

**Task Instruction:**
You will be provided with an answer and a reference for a specific question, based on its context. Your task is to rate the answer according to the reference provided.

**Evaluation Criteria:**
Factuality (1 - 5): Assess the accuracy and relevance of the answer based on how well it aligns with the reference. A high score indicates that the answer is factually correct and closely corresponds to the reference.

**Evaluation Steps:**
1. Understand the Reference: Carefully read the reference to identify its main topic and key points. 2. Compare the Answer: Review the provided answer and compare it against the reference. Evaluate whether the answer addresses the main topic and key points effectively, and whether it is presented in a clear and logical manner. 3. Assign a Score: Based on your comparison, rate the answer's factuality on a scale of 1 to 5, with 1 being the lowest and 5 being the highest, according to the guidelines below:

**Scoring Guidelines:**
- Score = 5: The answer fully captures all key points of the reference with accurate and logical flow, without significant omissions or irrelevant information.
- Score = 4 - 4.9: Most key points are included with a generally logical sequence, though there may be minor omissions or slight inclusions of less relevant information.
- Score = 3 - 3.9: Some key points are present, but others are missing, with noticeable gaps or jumps in the flow, and some irrelevant details.
- Score = 2 - 2.9: Several key points are missed, with a disjointed flow, significant omissions, inaccuracies, and noticeable irrelevant content.
- Score = 1 - 1.9: The answer fails to represent the reference accurately, lacks coherence and logical flow, with major elements missing or misrepresented, and significant irrelevant details.

**Context:**
[Your Context Here]

**Question:**
[Your Question Here]

**Reference:**
[Your Reference Here]

**Answer:**
[Your Answer Here]

**Evaluation Form:**
Begin your evaluation with "Analysis:" where you will concisely analyze the given answer according to the evaluation criteria. Then, provide your numeric rating on the next line starting with "Rating: [your rating here]. For example, "Rating: 4.2" without any other symbols or words.

---

Table 22: Instruction for reviewers in REFER

2052
2053
2054
2055
2056
2057
2058

*Instruction:*

**Task Instruction:**
You will be given an answer, a reference, and three evaluations from large language models for a specific question based on its context. Your task is to assess the accuracy of the answer by rating it according to the reference provided.

**Evaluation Criteria:**
Factuality (1 - 5): Assess the accuracy and relevance of the answer based on how well it aligns with the reference. A high score indicates that the answer is factually correct and closely corresponds to the reference.

**Evaluation Steps:**
1. Understand the Reference: Carefully read the reference to identify its main topic and key points. 2. Compare the Answer: Review the provided answer and compare it against the reference. Evaluate whether the answer addresses the main topic and key points effectively, and whether it is presented in a clear and logical manner. 3. Assign a Score: Based on your comparison, rate the answer's factuality on a scale of 1 to 5, with 1 being the lowest and 5 being the highest, according to the guidelines below:

**Scoring Guidelines:**
- Score = 5: The answer fully captures all key points of the reference with accurate and logical flow, without significant omissions or irrelevant information.
- Score = 4 - 4.9: Most key points are included with a generally logical sequence, though there may be minor omissions or slight inclusions of less relevant information.
- Score = 3 - 3.9: Some key points are present, but others are missing, with noticeable gaps or jumps in the flow, and some irrelevant details.
- Score = 2 - 2.9: Several key points are missed, with a disjointed flow, significant omissions, inaccuracies, and noticeable irrelevant content.
- Score = 1 - 1.9: The answer fails to represent the reference accurately, lacks coherence and logical flow, with major elements missing or misrepresented, and significant irrelevant details.

**Context:**
[Your Context Here]

**Question:**
[Your Question Here]

**Reference:**
[Your Reference Here]

**Answer:**
[Your Answer Here]

**First Assistant's Evaluation:**
[Peer response1]

**Second Assistant's Evaluation:**
[Peer response2]

**Third Assistant's Evaluation:**
[Peer response3]

**Evaluation Form:**
Begin your evaluation with "Analysis:" where you will concisely analyze the given answer according to the evaluation criteria. Then, provide your numeric rating on the next line starting with "Rating: [your rating here]". For example, "Rating: 4.2" without any other symbols or words.

Table 23: Instruction for area chair in REFER
