# OpenReview forum: "CALF: Benchmarking Evaluation of LFQA Using Chinese Examinations"
_ICLR.cc/2025/Conference — ICLR 2025 Conference Withdrawn Submission_

### Official Review · Reviewer_QA3V · 2024-10-31

**Soundness:** 3
**Presentation:** 2
**Contribution:** 2
**Rating:** 3
**Confidence:** 3

**Summary:**

The paper introduces a new benchmark, CALF, specifically designed to assess the evaluation metrics in Long-Form Question Answering (LFQA).  CALF is a reference-based benchmark derived from Chinese examination questions that have been translated into English. The dataset includes 1,476 examples with knowledge-intensive responses/references across six subjects. The authors design 3 experimental settings for a comprehensive analysis of 13 LFQA evaluation metrics,including traditional, prompt-based, and trained evaluation metrics. Extensive experiments reveal that current automatic metrics do not perform at human-level effectiveness.

**Strengths:**

1. The authors propose a relatively large and diverse dataset specifically designed to assess evaluation metrics for Long-Form Question Answering (LFQA), providing a valuable resource for the research community.

2. Three experimental settings facilitate a thorough analysis of 13 LFQA evaluation metrics, including 7 traditional, 3 prompt-based, and 3 trained metrics.

3. Extensive experiments reveal that current automatic metrics fall short of human-level performance, underscoring a significant gap in their ability to capture the nuanced information within long-form responses.

**Weaknesses:**

The organization of the paper could be improved, and there too many typos. The authors should carefully revise the writing to enhance clarity and readability.

**Questions:**

1. Given the numerous student responses for each question, what criteria were used to select response pairs, and how many pairs were ultimately chosen for each question?

2. In Table 2, the inter-annotator agreement is highest in the human vs. model setting, while in Table 4, most evaluation metrics achieve their best performance in the human vs. human setting. Why?

---

### Official Review · Reviewer_svhF · 2024-11-03

**Soundness:** 1
**Presentation:** 2
**Contribution:** 1
**Rating:** 3
**Confidence:** 2

**Summary:**

The paper addresses the challenge of evaluating LFQA and proposes a benchmark called CALF, derived from Chinese examinations and translated for evaluation
Existing LFQA evaluation lacks a standardized, effective benchmark, while CALF introduces a structured approach by leveraging question-answer data from authoritative Chinese examinations, creating an evaluation benchmark across diverse subjects. The authors evaluate CALF using both traditional and advanced evaluation metrics.

**Strengths:**

- CALF stands out by sourcing data from real, expert-written examination questions, providing high-quality, knowledge-rich content.

- The paper examines three evaluation settings to analyze the consistency and reliability of various metrics. However, these settings have been explored in previous works, which somewhat limits the novelty of this paper.

- The authors conduct a thorough analysis of traditional, prompt-only, and trained evaluation metrics, highlighting limitations and areas for improvement.

**Weaknesses:**

- Lack of Novelty in Addressing Evaluation Challenges: While the paper emphasizes the need for automatic, reliable, and human-aligned evaluation metrics for LFQA, it mainly explores current methods, demonstrating that even advanced LLMs struggle with LFQA’s subtlety requirements. Despite this challenge being highlighted in the introduction, the paper does not provide a clear path forward for addressing it, limiting its novelty.

- The CALF benchmark is based on Chinese examinations and annotations by workers from Online Marking Platforms (OMP). However, the paper has zero details on the annotators and data sources, raising concerns about the soundness of the benchmark, particularly in terms of annotation quality, data security and copyright and licensing of data sources.

- The description of the CALF benchmark is highly limited in the main text. Readers would benefit from additional details on the benchmark, including sample questions, answers generated by LLMs and annotators, and other characteristics that justify CALF’s validity for LFQA. While the appendix offers some category information and examples, more data is needed, such as question and reference lengths, topics, and question difficulty.

- Although using translated Chinese exam questions adds authenticity, it may introduce cultural or linguistic biases, potentially limiting CALF’s applicability in non-Chinese contexts. Specifically, the model-generated answers are used for law, psychology, and medicine topics—fields that vary greatly across cultures and languages. This factor could affect the generalizability of CALF in different linguistic settings.

- Lack of Related Works:  In section2, the authors rarely discuss advanced LLM-based LFQA evaluation methods, despite these being among their key baselines. Additionally, while G-Eval is mentioned early on (line 102), it isn’t appropriately cited until six pages later.

**Questions:**

See weakness

---

### Official Review · Reviewer_6iu7 · 2024-11-04

**Soundness:** 2
**Presentation:** 3
**Contribution:** 3
**Rating:** 5
**Confidence:** 4

**Summary:**

This paper presents CALF, a benchmark for evaluating LFQA metrics using translated Chinese exam questions. The approach leverages exam questions as reference material and evaluates various metrics under different comparison settings (human vs human, human vs model, model vs model). The experiments are comprehensive and covers 13 different evaluation metrics.

**Strengths:**

* three-way comparison framework is interesting and well-designed
* comprehensive evaluation across multiple metric types and models
* error analysis with examples is thorough

**Weaknesses:**

* why using translated content? Either focusing on original language of the dataset or focusing on some multilingual settings would be of higher value. Using machine translation to translate the dataset would add another layer of complexity and make the evaluation benchmark potentially more biased and less credible.
* no clear solutions to identified problems
* is there a need for all three evaluation settings? what can we get from these three different settings?

**Questions:**

save as above

---

### Official Review · Reviewer_b8vf · 2024-11-07

**Soundness:** 3
**Presentation:** 3
**Contribution:** 2
**Rating:** 5
**Confidence:** 3

**Summary:**

The paper introduces the Chinese Examination for Long Form QA Evaluation (CALF), a benchmark designed to evaluate the performance of automatic evaluation metrics for Long-Form Question Answering (LFQA). The authors address the challenge of assessing paragraph-level responses to open-ended questions, which is a complex task due to the depth and nuance required. CALF consists of 1476 examples from six diverse subjects, translated from Chinese to English, and includes both human-written and model-generated responses. The benchmark incorporates three comparison modes: human vs. human, human vs. model, and model vs. model. The paper presents an extensive experimental analysis of various evaluation metrics, including traditional, prompt-based, and trained metrics, against human judgment. The results indicate that current automatic evaluation metrics fall short of human performance, highlighting the need for more advanced evaluation systems. The main contribution is the proposal of a new dataset and the experiments results.

**Strengths:**

- The paper studies an important problem of evaluation, that whether we can adopt LLM to evaluate more complex questions regrading long response in the answer, which is the LFQA problem.
- The CALF benchmark covers a wide range of subjects. The inclusion of examples from geography, history, politics, law, medicine, and psychology ensures that the benchmark is not limited to a single domain.
- The conclusion is interesting that the newest evaluation metric involving LLMs can't beat some simple n-gram-based methods in the evaluation agreement with human experts.

**Weaknesses:**

- I believe that in addressing the LFQA evaluaiton problem, it is more crucial to diagnose the answers and provide detailed explanations for their scoring, rather than merely selecting which answer is better. This approach holds greater significance and practical application compared to simply ranking answers. Unfortunately, the proposed benchmark does not accomplish this task, which limits the contribution of the dataset.
- Why does LLM-based evaluation fall short in addressing the LFQA evaluation problem, even comparing to the simplest baseline like ROUGE/BLEU? The paper may offer valuable insights into developing improved evaluation systems. Additionally, in Tables 3 and 4, incorporating random guessing as a baseline could provide a clearer perspective on the absolute performance of the different evaluation metrics.
- I think the introduction should involve some examples to illustrate the evaluation pipeline and the cases, to let readers understand why evaluation of LFQA is a difficult problem.

**Questions:**

1. If the questions and answers originate in Chinese, why translate them into English for comparison? This process could lead to significant misalignment between the original meanings and the translated text, especially given that topics such as law, history, and politics are deeply rooted in specific ethnic and regional contexts.

---

### Note · Authors · 2024-11-22

I have read and agree with the venue's withdrawal policy on behalf of myself and my co-authors.